# A maize epimerase modulates cell wall synthesis and glycosylation during stomatal morphogenesis

Yusen Zhou [1,2,8], Tian Zhang[1,8], Xiaocui Wang [3,4,8], Wenqiang Wu [1,8], Jingjing Xing[1], Zuliang Li[1], Xin Qiao[1], Chunrui Zhang[3,4], Xiaohang Wang[1], Guangshun Wang[1], Wenhui Li[2], Shenglong Bai[1], Zhi Li[1], Yuanzhen Suo[5], Jiajia Wang[6], Yanli Niu[1], Junli Zhang[1], Chen Lan[1], Zhubing Hu[1,2], Baozhu Li[1], Xuebin Zhang [1], Wei Wang [1], David W. Galbraith [1,7], Yuhang Chen [3,4], Siyi Guo [1,2] ✉ & Chun-Peng Song [1,2] ✉

The unique dumbbell-shape of grass guard cells (GCs) is controlled by their cell walls which enable their rapid responses to the environment. The molecular mechanisms regulating the synthesis and assembly of GC walls are as yet unknown. Here we have identified *BZU3*, a maize gene encoding UDP-glucose 4-epimerase that regulates the supply of UDP-glucose during GC wall synthesis. The *BZU3* mutation leads to significant decreases in cellular UDP-glucose levels. Immunofluorescence intensities reporting levels of cellulose and mixed-linkage glucans are reduced in the GCs, resulting in impaired local wall thickening. BZU3 also catalyzes the epimerization of UDP-N-acetylgalactosamine to UDP-N-acetylglucosamine, and the *BZU3* mutation affects N-glycosylation of proteins that may be involved in cell wall synthesis and signaling. Our results suggest that the spatiotemporal modulation of BZU3 plays a dual role in controlling cell wall synthesis and glycosylation via controlling UDP-glucose/N-acetylglucosamine homeostasis during stomatal morphogenesis. These findings provide insights into the mechanisms controlling formation of the unique morphology of grass stomata.

Plant cell walls provide mechanical strength and modulate the directionality of cell growth, thereby determining cell shape and influencing their functions[1]. The dynamic cell wall structure plays a key role in modulating plant growth and development and also responses to biotic and abiotic stresses[2–4]. Wall formation involves synthesis, assembly, modification, and reorganization of different classes of wall polysaccharides and proteins, a process that is precisely coordinated and controlled[5,6].

Stomata are an excellent model for studying how alterations of cell walls modulate coordinated cell movement in response to

[1]State Key Laboratory of Crop Stress Adaptation and Improvement, School of Life Sciences, Henan University, Jinming avenue 1, Kaifeng 475004, China. [2]Sanya Institute, Henan University, Sanya 572025, China. [3]State Key Laboratory of Molecular Developmental Biology, Institute of Genetics and Developmental Biology, Chinese Academy of Sciences, Beijing, China; Innovative Academy of Seed Design, Chinese Academy of Sciences, Beijing 100101, China. [4]College of Advanced Agricultural Sciences, University of Chinese Academy of Sciences, Beijing 100049, China. [5]Biomedical Pioneering Innovation Center, School of Life Sciences, Beijing Advanced Innovation Center for Genomics, Peking University, Beijing 100871, China. [6]Joint National Laboratory for Antibody Drug Engineering, Henan University, Kaifeng 475004, China. [7]School of Plant Sciences and Bio5 Institute, The University of Arizona, Tucson, AZ 85721, USA. [8]These authors contributed equally: Yusen Zhou, Tian Zhang, Xiaocui Wang, Wenqiang Wu. ✉e-mail: guosiyi@henu.edu.cn; songcp@henu.edu.cn

environmental changes. Stomata provide the "gateway" for gas exchange between plants and their environment, regulating photosynthesis and transpiration and thereby influencing global plant carbon and water cycles[7,8]. Unlike other epidermal cells, the unique functions of stomata require substantial and repeated wall extension during their lifetime. The mechanical strength of the stomatal wall needs to withstand the high turgor pressure (0.7–5 MPa)[9,10], much larger than that of other plant cells (0.1–1 MPa). How stomatal walls are strong yet reversibly extensible, ensuring correct stomatal formation and precise movement, is a fascinating research question in the field. Compared to the kidney-shaped guard cells typical of eudicots, the dumbbell-shaped graminoid guard cells (GCs) are flanked by two subsidiary cells (SCs)[11–13]. Their presence allows the stomatal aperture to open and close more rapidly than that of non-graminoid stomata, increasing gas exchange and water-use efficiencies that are characteristic of grasses[14–16].

In both the kidney-shaped dicot GCs and the dumbbell-shaped grass GCs, turgor pressure arising from water influx and wall mechanics drives volume changes of the GCs, while the uneven thickening of the GC cell walls and the arrangement of cellulose microfibrils within these walls act together to control the direction of GC deformation[17]. In grasses, the elongated median regions of the dumbbell-shaped GCs have locally thickened cell walls at the region of the central canal, whereas the two bulbous ends have thinner walls. Cellulose microfibrils within the central canal region are arranged parallel to the GC long axis, and radiate from the central canal to the bulbous ends[18–21]. When GC turgor rises, the bulbous ends expand and push the two GCs apart in the plane of the leaf epidermis, resulting in a stomatal opening. Grass stomatal development can be divided into six developmental stages, characterized by the sequential presence of epidermal progenitor cells, guard mother cells (GMCs), subsidiary mother cells (SMCs), subsidiary cells (SCs), young stomata (YS), and mature stomatal complexes[14,22,23]. It remains poorly understood how changes in the composition and structure of the stomatal walls give rise to the unique shape and functions of grass GCs.

In higher plants, it is generally established that cellulose microfibrils are extruded extracellularly by cellulose synthase complexes (CSCs) moving on the plasma membrane. In contrast, matrix polysaccharides (hemicelluloses and pectins) are synthesized by glycosyltransferases (GTs), and are secreted to cell walls via vesicular transport[5]. Cell wall synthesis requires coordinated activities of sugar transporters and GTs in the Golgi apparatus[24]. Activated nucleotide sugars are the fundamental building blocks for the synthesis of cell wall carbohydrate polymers, their amounts being regulated by the activities of nucleotide sugar interconversion enzymes (NSEs). In plants, the presence of eleven NSE families has been confirmed, providing up to 16 different nucleotide sugar donors[25,26]. NSE family members include the UDP-glucose 4-epimerase (UGE) family, the members of which catalyze the reversible interconversion of UDP-glucose (UDP-Glc) and UDP-galactose (UDP-Gal). The UGE family is highly conserved in protein sequence and structure[27]. UGEs in *Arabidopsis* function in cell wall synthesis, and are involved in vegetative and reproductive growth and development[28]. In rice, mutation of *OsUGE2* resulted in brittle leaves and culms[25]. It is yet unknown whether the activities of UGEs also affect stomatal wall synthesis.

To address this question, we screened for maize mutants having abnormal stomatal function and morphology (*bizui, bzu*) retrieved from an ethyl methanesulfonate (EMS) mutagenesis population, and from the maizeGDB mutagenesis database (https://www.maizegdb.org/)[23]. In this study, we identified *bizui3* (*bzu3*), a maize mutant having collapsed GC walls and defects in local wall thickening. *BZU3* encodes a UDP-Glc 4-epimerase and plays a key role in controlling cell wall synthesis during the morphogenesis of maize stomata. It catalyzes epimerization of UDP-N-acetylgalactosamine (UDP-GalNAc) to UDP-N-acetylglucosamine (UDP-GlcNAc), regulating N-glycosylation patterns

of many proteins that may be involved in cell wall synthesis and signaling pathways. Our results suggest a bifunctional role of maize epimerase in cell wall synthesis and glycosylation during stomatal morphogenesis.

## Results

### *bzu3* plants display abnormal GCs and severely impaired stomatal function

Among stomatal deficient mutants emerging from the screen, two recessive single mutants were identified that had a translucent leaf appearance and that would die after growing for about 21 days (Fig. 1a, Supplementary Fig. S1a, and Supplementary Tables S1, S2). These two mutants had similar abnormal GC phenotypes (Fig. 1b and Supplementary Fig. S1b), and were verified as allelic mutants (Supplementary Table S3). We named these mutants *bzu3-1* (5808E from the maizeGDB database), and *bzu3-2* (an EMS mutant in the Zheng58 background). Intriguingly, the guard cells in *bzu3* mutants can be divided into two classes. Most (91.46 ± 5.03%) of the guard cells in *bzu3-1* mutants were antenna-shaped, with the central canals having collapsed onto a line, leaving the remains of the two bulbous ends. A small percentage (8.54 ± 5.03%) of the guard cells were rod-shaped, but the local wall thickening at the central canals was still absent. The ratios of abnormal GCs were similar in mutant *bzu3-2* (93.41 ± 3.73% antenna-shaped, and 6.59 ± 3.73% rod-shaped) (Supplementary Fig. S1c). By contrast, we counted 100% dumbbell-shaped GCs in the wild-type (WT) plants (Fig. 1b, c and Supplementary Fig. S1b–d). No differences in stomatal density were apparent between WT and *bzu3-1*, Zheng58, and *bzu3-2* plants (Supplementary Fig. S1e). Since plant cell walls provide the prime constraint on cell shape, we examined the cell wall structure using transmission electron microscopy. In *bzu3-1* plants, the GC walls collapsed and were much thinner (Fig. 1d). No significant differences were observed in the wall thickening of vascular bundles in leaves (Supplementary Fig. S2), implying the changes in cell wall thickening in the mutant were GC-specific. To define the stage of stomatal development of GCs at which the deformation occurred, we examined all stages of GC development of the stomatal lineage cells in the leaf base. As compared to the WT, the stomatal lineage cells in the mutant appeared normal until the later stages of stomatal elongation. Formation of the SC and of the pore aperture was not affected in the mutant (Fig. 1e). To further analyze the *bzu3* phenotype, we tested stomatal conductances under different light conditions and levels of $CO_2$. For the wild-type plants, strong lighting and low $CO_2$ levels induced stomatal opening, whereas darkness and high $CO_2$ levels induced stomatal closing. In contrast, the stomata of *bzu3* mutants remained closed and did not respond to changes in lighting and $CO_2$ levels. Thus, the abnormal GCs of *bzu3* resulted in stomatal malfunctioning (Fig. 1f, g).

### *BZU3* encodes a UDP-Glc 4-epimerase that specifically functions during GC maturation

To identify the gene responsible for the *bzu3* phenotype, we generated parental populations between inbred line B73 and plants carrying the mutation. Because homozygous *bzu3* mutant plants are lethal, we employed heterozygous *bzu3-1+/-* populations for crossing to B73. Map-based cloning using $F_2$ segregation populations identified the mutation in the 60.67-60.77 M region on the long arm of Chromosome 1, a region containing only one gene (Zm00001d029151; Fig. 2a). Genomic PCR amplification revealed a long transposon insertion in the second intron of Zm00001d029151 of *bzu3-1* mutant plants (Fig. 2b and Supplementary Fig. S3a–c). With primers designed for several regions of Zm00001d029151, RT-qPCR failed to detect any measurable transcripts from the sequences following the site of transposon insertion (Supplementary Fig. S3b, d). Sanger sequencing of *bzu3-2* mutant plants revealed a 1-base pair deletion in the 7th exon of Zm00001d029151, the same gene identified as defective in *bzu3-1*

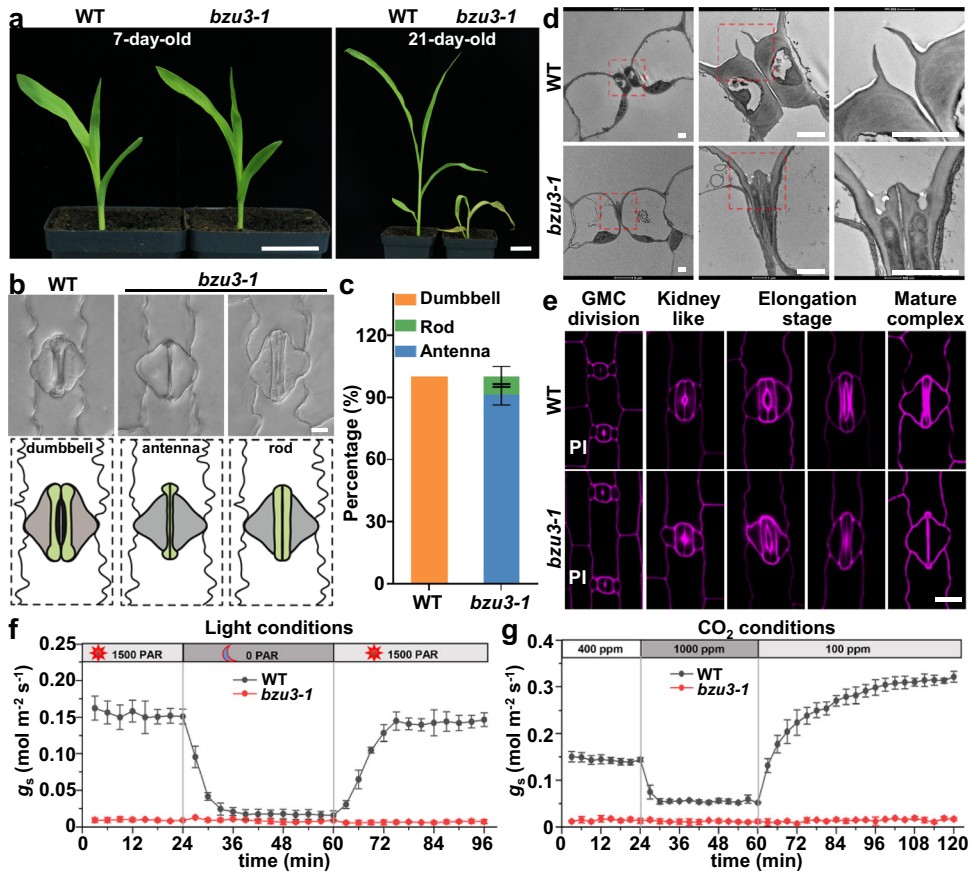

**Fig. 1 | Mutant *bzu3* displays abnormal guard cell shapes and is defective in local wall thickening. a** Representative images of 7- and 21-day-old seedlings of WT and *bzu3-1*. Scale bars, 3 cm. **b** Representative differential interference contrast (DIC) images (upper panel) and schematic diagram (lower panel) of stomata in WT, *bzu3-1* (the second leaf of 7-day-old seedlings). Scale bar, 10 μm. **c** Proportional phenotypic scoring analysis of stomatal types in WT and *bzu3-1*. The data was obtained by counting guard cells from ten seedlings of each genotype (WT: $n = 1104$, *bzu3-1*: $n = 1192$). Values represent means ± SD. SD, standard deviation. **d** Transmission electron microscope observation of stomata longitudinally sectioned of WT and *bzu3-1*. Red square dashed boxes indicate enlarged regions in the middle and rightmost panels, respectively. Scale bars, 2 μm. **e** Stomatal staining at different developmental stages in WT and *bzu3-1*. The cell wall was stained with propidium iodide (PI) and imaged via confocal microscopy. Scale bar, 20 μm. The images were obtained from an approximately 2-3 cm leaf base region taken from the third leaf of 7-day-old seedlings. **f** Stomatal conductance ($g_s$) in response to changing light conditions. Acclimation began at 1500 units of photosynthetically active radiation (PAR), dropped to 0 PAR, and then increased back to 1500 PAR. The data was obtained from three independent seedlings. Values represent means ± SD. **g** Stomatal conductance ($g_s$) in response to changing $CO_2$ level. Acclimation began at ambient $CO_2$ conditions (400 ppm), increased to 1000 ppm, and then dropped to 100 ppm. The data was obtained from three independent seedlings. Values represent means ± SD. Source data are provided as a Source Data file.

mutants (Fig. 2b and Supplementary Fig. S4a). The BZU3[bzu3-2] mutation results in a frame shift in the C terminus of the protein, and is predicted to severely impact protein stability (Supplementary Fig. S4b).

Three additional independent *bzu3* mutants (*bzu3-3*, *bzu3-4*, and *bzu3-5*) were generated in the B73-329 background using CRISPR-Cas9. All recapitulated the stomatal phenotype of *bzu3* mutants (Fig. 2b, c and Supplementary Fig. S5). To confirm the identity of the gene responsible for the phenotype, genetic complementation using the full-length CDS of Zm00001d029151 under its native promoter fully rescued the stomatal phenotype in *bzu3-2*, resulting in 100% dumbbell-shaped GCs, and also completely rescued the growth phenotypes of *bzu3-2* (Fig. 2d and Supplementary Figs. S6a, S7a). These results strongly support the proposition that the stomatal phenotype of *bzu3* mutant plants is due to defects in Zm00001d029151, which we renamed *BZU3*. BLAST searches and phylogenetic analyses indicated that *BZU3* encodes a UDP-glucose 4-epimerase (UGE) that shares homologs with *Arabidopsis thaliana*, rice and many other species across the plant kingdom (Supplementary Fig. S8).

To study the subcellular localization of BZU3, we performed transient expression of the BZU3-YFP fusion protein in maize mesophyll protoplasts. The results indicated that BZU3 was localized

to the cytoplasm (Supplementary Fig. S9). The expression pattern of *BZU3* in vivo was also observed in *BZU3pro:BZU3-YFP;bzu3-2* transgenic plants. During the early stages of stomatal development, *BZU3* was expressed in all epidermal cells. At later stages of stomatal development, the BZU3 signal gradually increased in GCs, being completely restricted to the GCs at the final stage of maturation (Fig. 2e and Supplementary Fig. S10). Statistical analysis of the BZU3-YFP signal indicated that the expression of *BZU3* notably increased from the point of GMC division to the stomatal elongation stage (Supplementary Fig. S10). To examine whether BZU3 functions specifically in GCs and to determine the precise stage of GC development at which BZU3 function is required, two stomatal-specific promoters, *BZU2/ZmMUTEpro* and *ZmFAMApro*[23], were used to drive *BZU3* for complementation testing. *ZmMUTEpro:BZU3-YFP* was expressed in GMCs and young guard cells until the elongation stage, but was barely detectable in mature GCs (Fig. 2f). *ZmFAMApro:YFP-BZU3* was expressed in GCs from the symmetrical division of GMCs to the point of GC maturation, but not during early stomatal development (Fig. 2g). The expression patterns of these two stomatal-specific promoters are temporally different but overlapping. *BZU3* driven by either promoter fully rescued the stomatal phenotype in *bzu3-2* (Supplementary Fig. S6b, c). Importantly, *ZmMUTEpro:BZU3-*

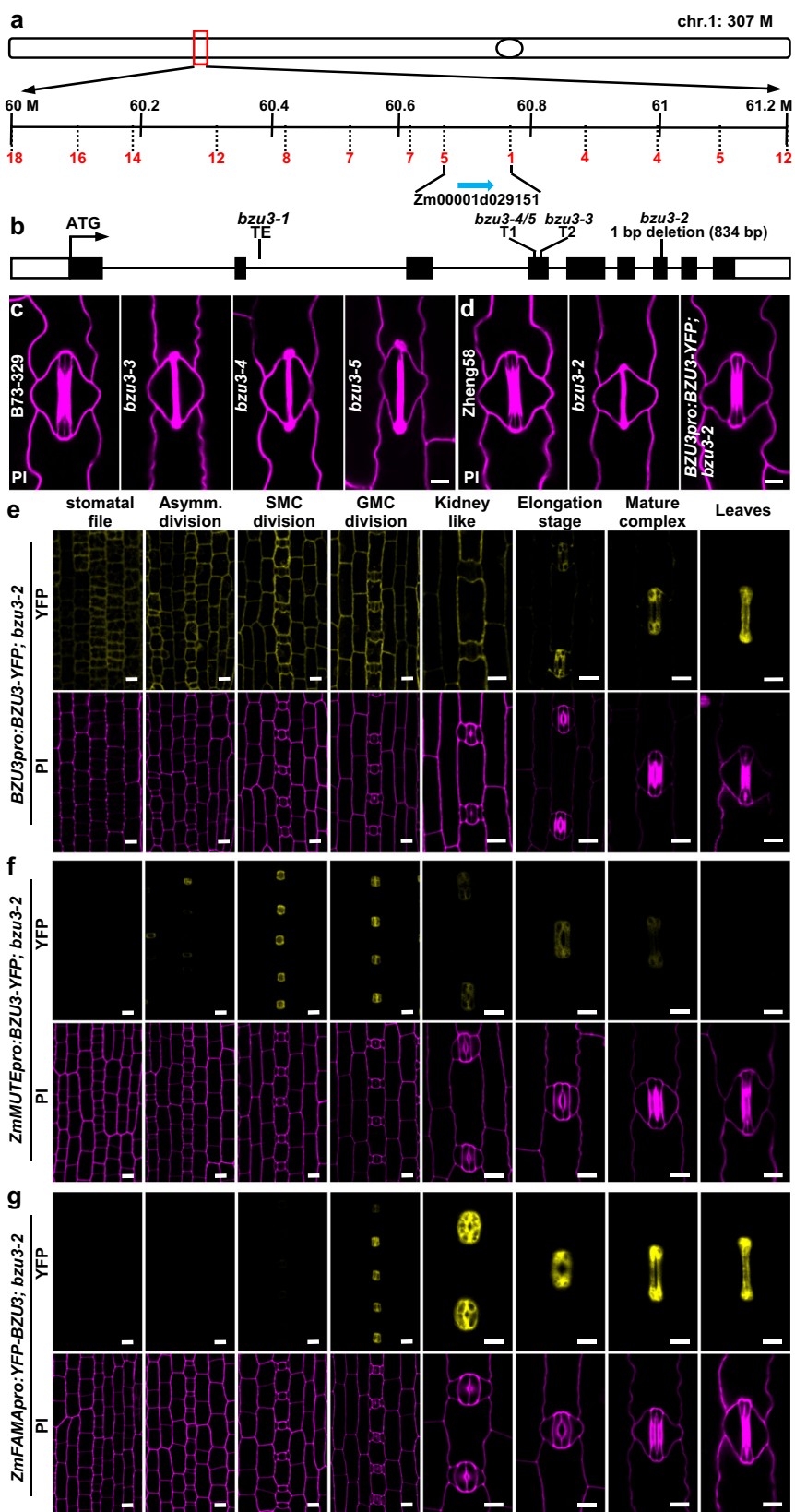

YFP and *ZmFAMApro:YFP-BZU3* could completely rescue the growth phenotypes of *bzu3-2* (Supplementary Fig. S7b, c). Considering that abnormal *bzu3* GCs appeared only at the later stage of stomatal elongation (Fig. 1e), our results suggest that BZU3 functions in GC morphogenesis during the specific time window at which the young guard cells elongate.

## BZU3 catalyzes interconversions of UDP-Gal/GalNAc and UDP-Glc/GlcNAc, affecting stomatal wall synthesis

Previous studies have shown that UDP-glucose 4-epimerases catalyze reversible conversions of UDP-Gal and UDP-Glc, and UDP-GalNAc and UDP-GlcNAc[26,29]. To characterize the enzymatic activity of BZU3, the recombinant protein was purified (Supplementary Fig. S11). With UDP-

**Fig. 2 | Identification and characterization of *BZU3* as the gene responsible for normal guard cell development in maize. a** Fine mapping and cloning of *BZU3* based on segregation from population *bzu3-1* crossed with B73. *BZU3* localizes to the region of bin 1.04 on the long arm of maize Chr 1. The fine-mapping region is 60.67-60.77 Mb and contains only one gene (Zm00001d029151). Chr, Chromosome. **b** Zm00001d029151 structure with 5′UTR, 9 exons, 8 introns, and 3′UTR. The transposon insertion, deletion mutation and CRISPR-Cas9 mutation positions are indicated. **c** Stomatal phenotype of B73-329, CRISPR-Cas9-produced mutations (*bzu3-3*, *bzu3-4*, and *bzu3-5*). **d** Stomatal phenotype of Zheng58, *bzu3-2*, and the *bzu3-2* line complemented by *BZU3pro:BZU3-YFP*. **e** *BZU3pro:BZU3-YFP* expression in *bzu3-2* during stomatal development. **f, g** *ZmMUTEpro:BZU3-YFP* and *ZmFAMApro:YFP-BZU3* remarkably complement homozygous *bzu3-2*. Representative confocal images of stomatal phenotypes of *ZmMUTEpro:BZU3-YFP* (**f**) and *ZmFAMApro:YFP-BZU3* (**g**) complemented *bzu3-2*. All images were observed from 7-day-old seedlings. YFP fluorescence was observed using confocal microscopy (Zeiss LSM710). Cell walls in the leaf epidermis were stained with PI (propidium iodide) and are indicated in purple. Scale bars, 10 μm.

Gal as substrate, the $K_m$ (Michaelis–Menten constant), $V_{max}$ (the maximum reaction velocity), and $K_{cat}/K_m$ were 1.45 mM, 1.41 mM·min$^{-1}$, and 650.3 s$^{-1}$·mM$^{-1}$ respectively. $K_m$, $V_{max}$, and $K_{cat}/K_m$ were 2.67 mM, 0.69 mM·min$^{-1}$, and 172.4 s$^{-1}$·mM$^{-1}$, respectively, when using UDP-Glc as substrate (Supplementary Fig. S12a, b and Supplementary Table S4). BZU3 also reversibly catalyzed the conversion of UDP-GalNAc to UDP-GlcNAc. $K_m$, $V_{max}$, and $K_{cat}/K_m$ were 0.50 mM, 0.36 mM·min$^{-1}$, and 474.7 s$^{-1}$·mM$^{-1}$, respectively, when using UDP-GalNAc as the substrate, whereas $K_m$, $V_{max}$, and $K_{cat}/K_m$ were 0.40 mM, 0.10 mM·min$^{-1}$, and 168.6 s$^{-1}$·mM$^{-1}$ with UDP-GlcNAc as the substrate (Supplementary Fig. S12c, d and Supplemental Table S4). BZU3 converted UDP-Gal/UDP-GalNAc to UDP-Glc/UDP-GlcNAc faster in the reversible isomerization reaction. When the reaction reached equilibrium, the ratio of UDP-Glc:UDP-Gal or UDP-GlcNAc:UDP-GalNAc was approximately 3:1 (Fig. 3a, b and Supplementary Fig. S13). Previous studies have shown that AtUGE1 and AtUGE3 can interconvert UDP-Xyl/UDP-L-Ara[30]. We employed a similar approach to test whether BZU3 interconverts UDP-Xyl/UDP-L-Ara. Our data indicate that whereas BZU3 can convert about 45% of UDP-Xyl to UDP-L-Ara, BZU3 was almost completely incapable of conversion of UDP-L-Ara to UDP-Xyl under our experimental conditions (Supplementary Fig. S14).

Since *bzu3* GCs showed defects in local wall thickening and BZU3 catalyzes the conversion of UDP-Gal to UDP-Glc, the main nucleotide sugar for cell wall synthesis, we examined different wall components present in maize stomata. Stomata at later stages of development were stained with Direct Red 23, a dye that specifically identifies cellulose[31,32]. In WT, the red fluorescent signal in GC walls gradually increased, indicating deposition of cellulose and GC wall thickening (Fig. 3c, d. $P < 0.001$). No gradual accumulation of cellulose signals was observed in *bzu3* plants. Immunofluorescence labeling suggested that the amount of mixed-linkage glucan (MLG) in the GC wall of *bzu3* was also significantly reduced as compared to the WT (Fig. 3e, f. $P < 0.001$). The cell wall composition of mature stomata was also analyzed using stimulated Raman scattering microscopy (SRS). The intensity of the Raman signals for cellulose and hemicellulose were significantly lower in GCs of *bzu3* than that of the WT (Fig. 3g, h. $P < 0.001$). Immunofluorescence labeling indicated that the amount of homogalacturonan in the GC wall was very low, and it was difficult to determine whether there was a difference between the wild type and the mutant (Supplementary Fig. S15). Collectively, these results suggest that severely impaired cellulose and MLG synthesis leads to defects in local wall thickening and collapse of maize GCs in *bzu3* plants.

**Structures of BZU3 docked with NAD⁺/UDP-Glc and NAD⁺/UDP-GlcNAc**

To determine whether BZU3 also catalyzes the conversion of UDP-Gal to UDP-Glc in vivo, we performed UPLC-MS analysis of nucleotide sugars in the maize leaf base, finding that levels of UDP-Gal were remarkably increased, and UDP-Glc levels reduced, in *bzu3* as compared to the WT (Fig. 4a. $P < 0.001$). This suggests that mutation of BZU3 results in a disturbed balance between the cellular concentrations of UDP-Gal and UDP-Glc. To test whether a reduced level of UDP-Glc led to the stomatal phenotype in *bzu3*, exogenous UDP-Glc was applied at the maize leaf base. Some stomata (~10%) recovered the normal dumbbell shape, indicating that UDP-Glc plays a crucial role in maize GC development (Fig. 4b).

To further address the molecular functions of BZU3, we solved the 3-D structures of the BZU3 protein, both in *apo* form and in complexes with NAD⁺/UDP-Glc and NAD⁺/UDP-GlcNAc, at resolutions of 2.6 Å, 1.25 Å, and 2.15 Å respectively (Supplementary Table S5). BZU3 formed a homo-dimer (Supplementary Fig. S16a, b), with each protomer folded into two distinct structural units, the N-terminal NAD⁺ binding domain (residues 1-189) and the C-terminal sugar-binding domain (residues 190-355). The catalytic site was located in the cleft between these two structural domains (Fig. 4c). The homo-dimerization structure was consistent with in vivo bimolecular fluorescence complementation (BiFC) results (Supplementary Fig. S16c). BZU3$^{bzu3-2}$ failed to form homo-dimers, probably due to the frameshift which would be expected to alter the structural folding of its C-terminal domain (Supplementary Fig. S16d). The flexible sugar moiety was held in place in the vicinity of the catalytic triad of Ser-Tyr-Lys (S133/Y157/K161). S133 forms a bifurcated contact with the sugar moiety of the substrate, whereas both Y157 and K161 directly interact with the ribose moiety of NAD⁺ (Fig. 4c). The maize BZU3 shared similar structures with the GALE proteins from humans and *E. coli* (Supplementary Fig. S17). The highly conserved NAD⁺ binding motifs (G$^{13}$XXG$^{16}$XXG$^{19}$ and Y$^{157}$XXXK$^{161}$) of the UGE family were indicated in the aligned sequences. The structure of BZU3 in complex with NAD⁺/UDP-GlcNAc was similar to that of BZU3/NAD⁺/UDP-Glc, except that the sugar moiety of the substrate adopts a distinct orientation (Fig. 4d).

Drawing on the structural model, we designed mutations within the conserved catalytic triad and found that S133A significantly reduced catalytic activity, while Y157F completely abolished the interconversion between UDP-Glc and UDP-Gal, and UDP-GlcNAc and UDP-GalNAc (Fig. 3a, b, Supplementary Fig. S13, and Supplementary Table S6). Consistent with this result, the transgene of *BZU3pro:BZU3$^{Y157F}$-YFP* failed to rescue the stomatal phenotype in *bzu3-2* (Fig. 4e and Supplementary Figs. S6d, S7d). Taken together, these observations imply that the catalytic activity of BZU3 is required for the morphogenesis of maize GCs.

**Mutation of BZU3 alters N-glycosylation pattern**

Previous studies have shown that UDP-GlcNAc is the main donor for N-glycosylation[33]. Considering that BZU3 also catalyzes UDP-GalNAc to UDP-GlcNAc in vitro, we undertook glycoproteomics analyses of the N-glycans. Based on the libraries of maize protein group and plant N-glycosylation modification in the Uniprot database, we identified 2,194 intact N-glycopeptides, and quantified 181 differentially expressed intact N-glycopeptides (DEGPs) (two out of the three technical replicates, ≥ 1.5-fold change and p-value < 0.05) (Supplementary Fig. S18). A total of 173 proteins decorated by all three N-glycan types were down-regulated in *bzu3-2*, while only 8 proteins decorated by the core N-glycan type were up-regulated (Fig. 5a). Consistent with this result, down-regulated N-glycan species in the mutants were also much more abundant as compared to their up-regulated counterparts. As expected, N-glycan species containing xylose and fucose were largely reduced in the mutants as compared to the wild-type (Fig. 5b). These

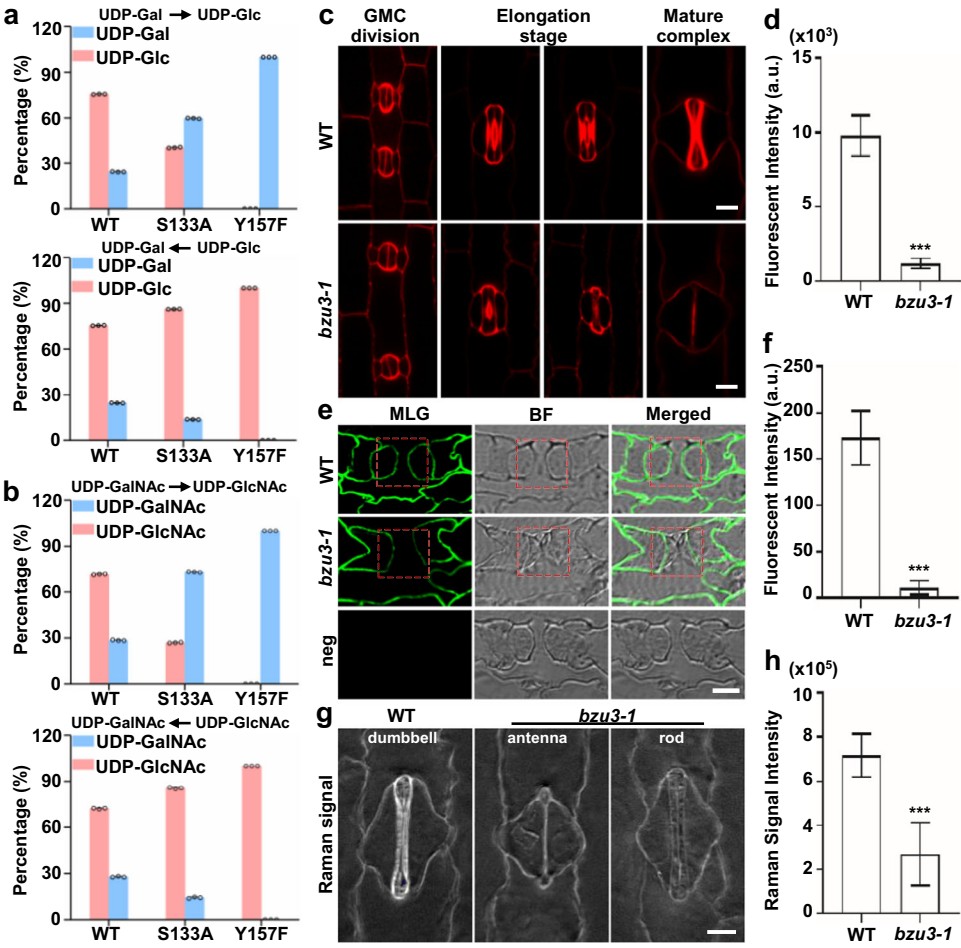

**Fig. 3 | BZU3 catalyzes conversion of UDP-Gal to UDP-Glc in vitro, and *bzu3* mutants show defective cell wall synthesis. a**, **b** Catalytic activity of BZU3 and its variants S133A and Y157F as determined by UDP-Glc, UDP-Gal, UDP-GlcNAc, and UDP-GalNAc. The data was obtained from three biological replicates. Values represent means ± SD. SD, standard deviation. **c** Representative confocal images of cellulose stained fluorescence of WT and *bzu3-1*. Cellulose from leaf base were stained using Direct Red 23, with fluorescence being observed using a Zeiss LSM710. Scale bars, 10 μm. **d** Statistical analysis of Direct Red 23 staining signal intensity in WT and *bzu3-1* mature guard cells (**c**). The signal intensity was obtained from three seedlings of each genotype (*n* = 32 guard cells). Values represent means ± SD. ***P = 4.998E-42 by Student's *t* test (two-sided). SD, standard deviation. **e** Representative images of MLGs immunolabeling with BG-1 antibody in resin-embedded sections (1.5 μm thick) of leaf base. Labeling was detected with a FITC-labeled secondary antibody and visualized by fluorescence microscopy. neg, negative control. Red square dashed boxes indicate guard cell. Scale bar, 5 μm. **f** Statistical analysis of MLGs immunolabeling signal shown in (**e**). The signal intensity was obtained by measuring guard cells from three seedlings of each genotype (*n* = 30). Values represent means ± SD. ***P = 2.365E-36 by Student's *t*-test (two-sided). SD, standard deviation. **g** Representative images of stimulated Raman scattering microscopy of the cell wall of the epidermis in WT and *bzu3-1*. The images were produced using Raman spectra with 1100 nm. Scale bar, 10 μm. **h** Statistical analysis of relative content of cellulose and hemicellulose in stomata of WT and *bzu3-1*. The signal intensity was obtained by measuring guard cells from five seedlings of each genotype (WT: *n* = 100, *bzu3-1*: *n* = 103) Value represents mean ± SD. ***P = 1.142E-65 by Student's *t*-test (two-sided). SD, standard deviation. Source data are provided as a Source Data file.

two monosaccharide modifications play important roles in a wide variety of biological processes, including immune responses, signal transduction, and cell development[34–36]. Gene ontology analysis revealed that these affected N-glycoproteins were enriched in categories related to metabolism and biosynthesis of cell walls, cell-surface receptors, signaling, etc (Fig. 5c). Mass spectrometric analysis of the differentially expressed proteins also identified many glycosyl-transferases and leucine-rich repeat receptor-like kinases (Supplementary Data S1). Within the differentially expressed proteins, e.g. PAN2 (PANGLOSS2), GT14 (Glycosyltransferase 14), and FUT (Fucosyltransferase, Zm00001d014505), RPLC-MS/MS analysis demonstrated that the N-glycosylation sites of PAN2/GT14/FUT were N554/N283/N121, respectively (Fig. 5d–f). Furthermore, we performed a gel shift assay of proteins generated by site-directed mutagenesis. PAN2^{N554Q}/GT14^{N283Q}/FUT^{N121Q} proteins migrated faster than PAN2/GT14/FUT proteins (Supplementary Fig. S19a–c), indicating that PAN2, GT14, and FUT can indeed be N-glycosylated.

## Discussion

In this study, we provide cellular and structural evidence that BZU3, a maize UDP-glucose 4-epimerase, plays a key role in modulating stomatal morphogenesis. Firstly, we demonstrated that BZU3 is responsible for supplying UDP-Glc for stomatal wall formation, thus ensuring normal stomatal function. Secondly, in vitro enzymatic studies and crystallography results indicate that BZU3 catalyzes the conversion of UDP-Gal/GalNAc to UDP-Glc/GlcNAc. Lastly, our glycomics data suggest that BZU3 affects N-glycosylation of many proteins that are involved in wall formation and signaling.

Grass stomata can respond to environmental changes more rapidly, which may contribute to their widespread and diversification during a global aridification 30-40 million years ago[13]. Their characteristic dumbbell shape, along with the uneven pattern of cell wall thickening, raises questions as to the building blocks required for graminoid stomatal wall synthesis, and how cell wall composition and structure modulate grass stomatal shape and function. Our

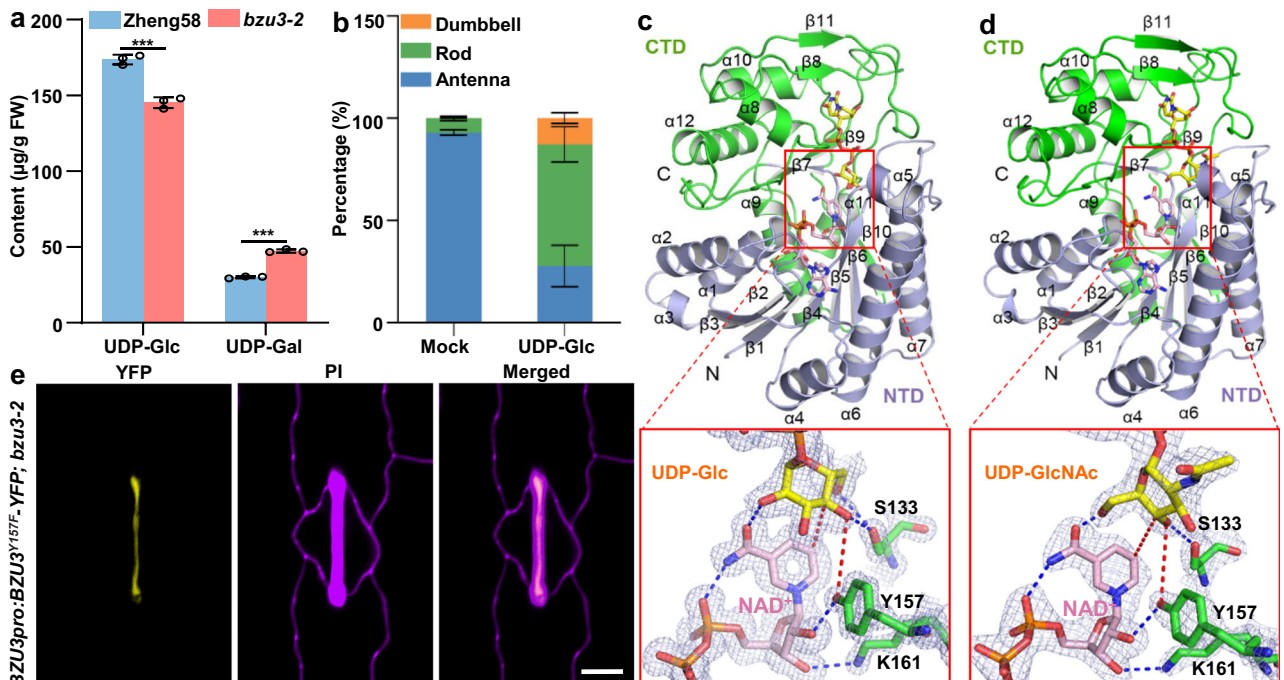

**Fig. 4 | BZU3 catalyzes UDP-Gal epimerization to UDP-Glc in vivo, and UDP-Glc is crucial for maintaining the dumbbell-shape of maize guard cells. a** UPLC-MS detection of UDP-Glc and UDP-Gal in Zheng58 and *bzu3-2*. 0.1 g leaf base of Zheng58 and *bzu3-2*. The results were performed three independent biological repeats with similar results. Values represent means ± SD. ***$P$ = 0.0005 of UDP-Glc and ***$P$ = 1.773E-05 of UDP-Gal by Student's *t*-test (two-sided). SD, standard deviation. **b** Phenotypic statistical analysis of *bzu3-2* treated with UDP-glucose. The data was obtained by counting guard cells from five seedlings of each genotype (Mock: $n$ = 2901, UDP-Glc: $n$ = 3283). Values represent means ± SD. SD, standard deviation. **c** Overall structure of BZU3/NAD⁺/UDP-Glc complex. Cropped view shows the interactions between BZU3, NAD⁺, and UDP-Glc at the reaction center. **d** Overall structure of BZU3/NAD⁺/UDP-GlcNAc complex. Cropped view shows the interactions between BZU3, NAD⁺, and UDP-GlcNAc at the reaction center. **e** *BZU3pro:BZU3^{Y157F}-YFP* failed to rescue the stomatal phenotype in *bzu3-2* mutant. The images were obtained from the second leaf of the 7-day-old seedlings. Propidium iodide (PI, purple) was used to stain cell walls. Scale bar, 10 μm. Source data are provided as a Source Data file.

biochemical data indicated that the epimerase activity of BZU3 prefers to catalyze conversion of UDP-Gal to UDP-Glc (Fig. 3). In *bzu3* mutants, local wall thickening at the central canal region of the dumbbell-shaped stomata is severely impaired (Fig. 1). Further analysis revealed reduced levels of UDP-Glc and reduced labeling of cellulose and mixed-linkage glucan in the stomatal wall (Fig. 3). These results suggest that BZU3 is the key player in providing UDP-Glc for maize stomatal wall formation, which is essential for stomatal morphogenesis and function. In most cell types, UDP-Glc is predominantly supplied to plasma membrane-associated CSCs through pathways involving sucrose synthases (SuSy) and UDP-glucose pyrophosphorylase (UGPase)[24,37]. The alternative or compensatory supply of UDP-Glc provided by BZU3 in maize guard cells suggests that these precisely spatiotemporal-regulated epimerase activities evolved to determine the special shapes and functions of maize stomates. Cell wall composition and structure, with stiff cellulose microfibrils embedded in a soft matrix, are suggested to be very important for normal stomatal functioning in many other plants, including *Arabidopsis, Commelina,* and mosses[17,19,38]. Defects in cellulose synthesis due to inhibition or mutation of CSCs or degradation of cellulose result in significantly larger stomatal apertures[39].

BZU3 belongs to the UGE superfamily, whose members are highly conserved in gene sequence and structure (Supplementary Fig. S8)[27]. In *Arabidopsis*, there are five UGEs that are suggested to affect vegetative growth and pollen development. No defects related to stomatal morphology were found in *atuge2, atuge4,* and *atuge2 atuge4* mutants (Supplementary Fig. S20)[28]. There are at least four putative UGEs in rice[28]. Previous studies have suggested that the function of OsUGE2 is related to cell wall synthesis[25], but no stomatal defects were observed in *osuge2* mutants (Supplementary Fig. S21). Our study in rice found

that mutation of *OsUGE1*, which has the closest phylogenetic relationship with *BZU3*, also did not result in stomatal defects (Supplementary Fig. S21). Recently, Wang et al. found that OsUGE1, as a transcriptional activation factor, is crucial for the degradation of tapetum and male fertility in rice[40]. Meanwhile, Yang et al. found that *OsUGE1* is negatively regulated in rice root hair elongation by targeted regulation of OsGRF6[41]. Intriguingly, mutagenesis of rice *OsUGE3* also led to abnormal stomata (antenna-shaped and rod-shaped) (Supplementary Fig. S21). Although there are four additional UGE homologs in maize, we identified only two GC UGEs (including BZU3) in the single-nucleus transcriptome[42]. Compared to *Arabidopsis* UGEs, BZU3 is specifically required at the later stage of stomatal elongation, a time when the GC wall becomes locally thickened at the central canal, a process that would require copious amounts of UDP-Glc to accommodate the increased activity in cell wall synthesis. The isomerization of UDP-Gal to UDP-Glc by BZU3 might conveniently satisfy this requirement. Considering the differences between grasses and dicots in stomatal shape, wall composition, and structure, the spatiotemporal-regulated epimerase activities of maize BZU3 may have evolved to accommodate the highly specialized shape and function of grass stomata.

Upon exposure to exogenous UDP-Glc, the dumbbell-shaped stomata of the *BZU3* mutants were not completely recovered. One simple explanation is that the exogenous supply of UDP-Glc was not sufficiently taken up by the cells, but since exogenous UDP-Glc was capable of restoring the growth phenotype of at least some UDP-Glc deficient plants, we examined other possibilities[43,44]. An alternative explanation for the partial recovery of the stomatal phenotype may be that BZU3 controls other regulatory events, such as post-translational modifications of proteins, during stomatal development.

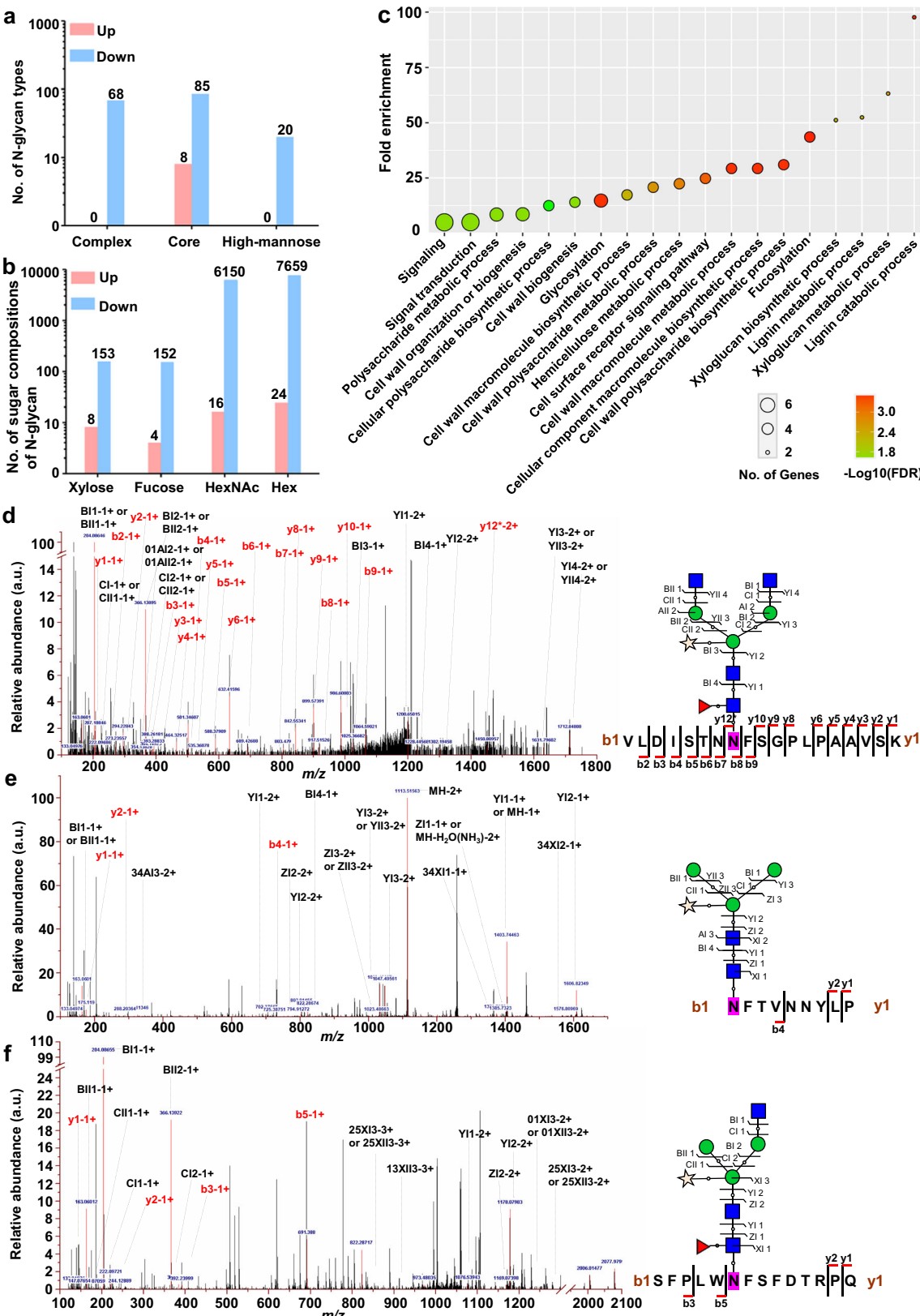

**Fig. 5 | Glycoproteomics analysis identifies altered glycosylation in *bzu3* mutants. a** The patterns of N-glycans are affected by *BZU3* mutation. **b** Monosaccharides composition of N-glycans is also affected by *BZU3* mutation. **c** Gene ontology enrichment of genes based on the glycoproteomics results. **d** Site-specific identification of N-glycopeptide PAN2_VLDISTNNFSGPLPAAVSK (01Y(31 F)41Y41M(31 M21Y)(21X)61 M21Y). **e** Site-specific identification of N-glycopeptide GT14_NFTVNNYLR (01Y41Y41M(31M)(21X)61M). **f** Site-specific identification of N-glycopeptide FUT_SFPLWNFSFDTRPQ (01Y(31 F)41Y41M(31 M)(21X)61M21Y). Blue square: Y, GlcNAc; Green circle: M, Mannose; Red triangle: F, Fucose; Orange pentagon: X, Xylose. PAN2: Zm00001d007862, GT14: Zm00001d008513, FUT: Zm00001d014505. Source data are provided as a Source Data file.

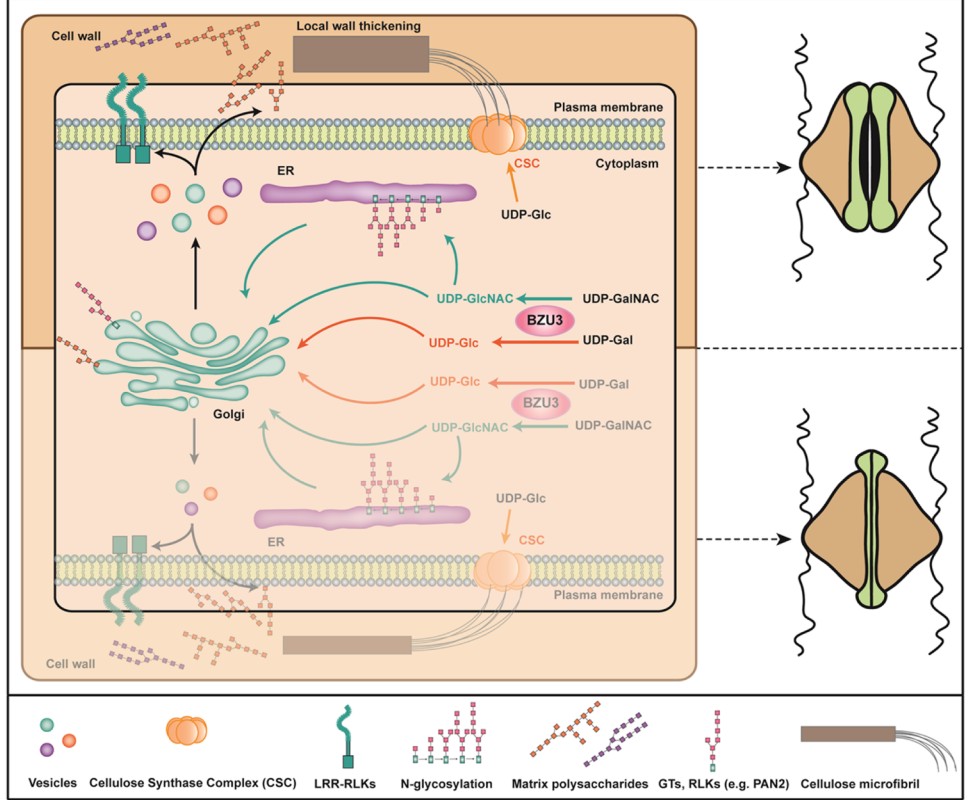

**Fig. 6 | A proposed working model for BZU3.** BZU3 regulates the balance between UDP-Glc and UDP-Gal in maize GCs. The nucleotide sugars are building blocks for the biosynthesis of cellulose and matrix polysaccharides in plant cell wall. BZU3 also catalyzes UDP-GalNAc epimerization to UDP-GlcNAc, the important donor for post-translational modification of protein N-glycosylation. Mutation in *BZU3* causes altered glycosylation patterns of many cell wall-related proteins such as glycosyl-transferases (GTs) and leucine-rich repeat receptor-like kinases (LRR-RLKs). Thus, BZU3 plays a key role in regulating cell wall synthesis and glycosylation during the morphogenesis of maize GCs.

Post-translational modification, for example, N-glycosylation, ensures correct folding and normal functions of proteins. Thus, an alternative role of BZU3 may relate to its catalytic activity regulating the level of UDP-GlcNAc, an important donor for N-glycosylation. Indeed, crystallized BZU3 protein bound to NAD⁺/UDP-GlcNAc (Fig. 4), and our glycoproteomics data identified many proteins whose glycosylation patterns were affected by BZU3 mutation, including glycosyltransferases (GT14 and FUT), leucine-rich receptors PAN2, and wall-associated kinases as suggested by GO analysis (Fig. 5 and Supplementary Data S1). Therefore, an alternative hypothesis is that a key developmental step prior to extensive cell wall modification, maturation, and thickening are affected.

The glycosylation site of PAN2 at N554 was further confirmed by gel-shift assays (Supplementary Fig. S19a). We hypothesized that, similar to the *pan2* mutant[45], *bzu3* mutants with abnormal glycosylation patterns of PAN2 may show similar phenotypes in the shape of subsidiary cells and inter-stomatal cells. As shown in Supplementary Fig. S19d, a typical triangular subsidiary cell has the following wall segments, one is the inter-stomatal cell shared with subsidiary cells (marked in yellow and named as x), and the other one is flanking non-stomatal row cells which form one of three points becoming the triangular subsidiary shape (marked in red and named as y). The *bzu3* mutant subsidiary cell shapes differ slightly from the wild-type (Zheng58), with a significantly shorter wall (x) of subsidiary cells in *bzu3-2* (14.9 ± 0.1 μm) compared with the wall segments of wild-type (16.4 ± 0.3 μm) (Supplementary Fig. S19e). By contrast, the length of wall y in the *bzu3-2* mutant was not significantly different from Zheng58 (13.2 ± 0.1 μm versus 13.4 ± 0.4 μm) (Supplementary Fig. S19f). These observations showed similar a phenotype in the shape

of subsidiary cells in *bzu3* and *pan2* mutants, implying that the glycosylation of PAN2 was altered in *bzu3* which, in affecting the function of PAN2, results in a phenotype in *bzu3* that is partially similar to the shape of subsidiary cells seen with *pan2*. It warrants further investigation to provide details concerning the regulatory role of BZU3 during stomatal morphogenesis via signaling and other mechanisms that are uncovered.

In summary, our results demonstrate a key role for BZU3, as an epimerase, in modulating the level of UDP-Glc for maintaining the dumbbell shape of maize GCs. In addition, BZU3 also influences the N-glycosylation patterns of many glycosyltransferases and cell-surface receptors that may be involved in the synthesis of stomatal wall matrix or signaling events during stomatal morphogenesis. Thus, the mutation of *BZU3* leads to defects in stomatal walls and affects the morphogenesis and functions of maize GCs (Fig. 6). Since the unique dumbbell-shaped grass stomata are more efficient in regulating gas exchange, the genetic basis and regulatory mechanisms of grass stomatal morphogenesis will provide valuable insights for improving the efficiency of stomata and facilitate targeted-molecular breeding in a changing climate.

## Methods

### Plant materials and growth conditions
The *bzu3-1* mutant in the 5808E background was obtained from the Maize Genetics Cooperation Stock Center (http://maizecoop.cropsci.uiuc.edu/). The *bzu3-2* mutant in the Zheng58 background was isolated from an EMS (ethyl methanesulfonate) mutagenized database. Both *bzu3-1* and *bzu3-2* mutants were identified from self-crossed heterozygotes. Seedlings of WT, *bzu3-1*, Zheng58, and *bzu3-2* were planted in

the greenhouse at 16 h/8 h and 32 °C/25 °C (day/night). For all field trials, WT, *bzu3-1+/-*, Zheng58, *bzu3-2+/-* plants were planted in Kaifeng, Henan in summer, and in Sanya, Hainan in winter.

## Stomatal phenotyping by microscopy

For stomatal density and developmental stage analyses, the second leaf of 7-day-old seedlings was collected and treated overnight with ethanol/acetic acid solution (7:1, v/v). The stomata on the abaxial side of the leaf epidermis were observed using an Olympus CX43 microscope. Under the 10× objective, we counted stomata from each leaf of 10 plants (5–7 vision fields per leaf). The visual field under the 10× objective was fully occupied by the leaf epidermis, and its area was 1.142 mm$^2$. The stomatal density was calculated as the number of stomata per area of the visual field. Different types of stomata were identified using the 20× objective, and we counted ~1000 stomata from 10 plants. To observe the lineages of the stomatal cells throughout stomatal development, young leaf segments taken from the bottom 2–3 cm of the third leaf of 7-day-old seedlings were cut into 0.2 cm × 0.5 cm pieces and stained for 5 minutes with 0.01 mg/mL propidium iodide (PI). After rinsing with ddH$_2$O, the samples were observed by a Zeiss LSM710 confocal microscope with excitation at 561 nm.

Transmission electron microscopy was mainly based on Yadav et al. with slight modifications[46]. The middle part of the second leaf was collected from the 7-day-old seedlings. The leaf samples were fixed immediately in 5% glutaraldehyde and 4% paraformaldehyde buffered with 0.1 M sodium phosphate, pH 7.2. After being rinsed in the same buffer, the samples were postfixed with 1% osmium tetroxide in 0.1 M sodium phosphate, pH 7.2 for 3 h at 4 °C. Samples were dehydrated in an ethanol gradient series, then transferred to acetone, and finally infiltrated and embedded with Epon812 epoxy resin. Ultra-thin sections were obtained using a Leica EM UC7 ultramicrotome, and were stained with uranyl acetate and lead citrate. The sections were observed under a transmission electron microscope (FEI, TECNAI G2 Spirit Bio TWIN).

## Stomatal conductivity

Seedlings were grown to the 3-leaf stage, the second leaf being used to measure stomatal conductivity with a Li-COR LI-6800 photosynthetic apparatus. The parameter settings were: (1) Flow: On; Pump Speed: Auto; Flow Setpoint: 500 μmol s$^{-1}$; Press. Valve: 0.1 kPa. (2) H$_2$O: On; RH_air: 55%. (3) CO$_2$ injector: On; CO$_2$_s: 400 μmol mol$^{-1}$ (adjusted according to experimental requirements). (4) Mixing fan: On; Fan Speed: 10,000 rpm. (5) Temperature: On; Tleaf: 25 °C. (6) Light: 1,500 μmol m$^{-2}$ s$^{-1}$ (adjusted according to experimental requirements). (7) Select Auto Log program: 3 min acquisition interval, and the total acquisition time is 2-3 h.

## Map-based cloning for *bzu3*

Since the mutation is lethal in homozygotes, we used *bzu3-1+/-* × B73 to generate parental populations for map-based cloning. For preliminary mapping of *bzu3-1*, we employed a total of 384 SSR markers (SIGMA catalog no. M4193) selected from the maize genetic and genomic database. The entire genome was covered with an average of 20 cM units of mapping distance between two SSR markers. The mutation of *bzu3-1* was localized to the long arm of the 1$^{st}$ chromosome. Next, 60 pairs of SSR markers covering 20–110 M of chromosome 1 were employed, and *bzu3-1* was localized to 53.7–63.7 M. 7000 F$_2$ plants were used for fine mapping, and finally, *bzu3-1* was localized at the region of 60.67–60.77 M on Chromosome 1. Searching the MaizeGDB database revealed only one gene (Zm00001d029151) in this region. With Sanger sequencing, we found that a viral retrotransposon, having a full length of 5–6 kb and containing forward long terminal repeats, was inserted into the second intron in Zm00001d029151 in *bzu3-1* mutant plants.

## Generation of CRISPR lines

To generate the CRISPR-Cas9 mutant of *BZU3*, the sgRNAs of BZU3 were designed and evaluated using the CRISPOR online server (http://crispor.tefor.net/). The sgRNAs were inserted into the pBUE411 vector and verified by Sanger sequencing. The pBUE411-*BZU3*-sgRNA1-sgRNA2 construction was transformed via agrobacteria (strain EHA105) into inbred line B73-329 as previously described[23]. Positive transgenic lines were confirmed by PCR and verified by sequencing. Three stably-inherited alleles were selected for further phenotypic investigation. The *BZU3* CRISPR-Cas9 mutants were produced in the Plant Transformation Facility, at Henan University. The primers were listed in Supplementary Table S7.

## Genetic complementation

The promoter sequences of *BZU3*, *ZmMUTE*, and *ZmFAMA* were amplified from the maize genome, and the *BZU3* sequence was amplified from cDNAs. We constructed *BZU3pro:BZU3-YFP*, *ZmMUTE-pro:BZU3-YFP*, and *ZmFAMApro:YFP-BZU3* vectors. Vector *BZU3pro:BZU3$^{Y157F}$-YFP* was constructed using *BZU3pro:BZU3-YFP* as a template. Maize plants of the inbred line B73-329 were transformed using agrobacteria EHA105 carrying the vectors described above. *BZU3pro:BZU3-YFP* transgenic plants were crossed with *bzu3-2+/-* to produce the F$_1$ generation. F$_2$ plants were segregated from self-crossed *BZU3pro:BZU3-YFP* (♂) × *bzu3-2+/-* (♀) F$_1$ generation and *BZU3pro:BZU3-YFP;bzu3-2* plants were isolated. Similarly, *ZmMUTEpro:BZU3-YFP;bzu3-2*, *ZmFAMApro:YFP-BZU3;bzu3-2*, and *BZU3pro:BZU3$^{Y157F}$-YFP;bzu3-2* plants were obtained. The stomatal phenotype of this complemented transgenic maize was observed and compared to the WT. The primers were listed in Supplementary Table S7.

## Purification of BZU3 and enzyme kinetics analyses of BZU3

The coding sequence of *BZU3* was cloned into the pET28a vector to construct *T7pro:6×His-BZU3*, from which *BZU3$^{S133A}$* and *BZU3$^{Y157F}$* were made using site-directed mutagenesis. The primers were listed in Supplementary Table S7. The plasmids were transformed into *E.coli* Rosetta (DE3), and proteins were expressed following 0.3 mM IPTG induction at 23 °C for 16 h. The cells were collected and resuspended in 1×PBS buffer containing 1 mM PMSF. After cell lysis, proteins were purified on a nickel column and examined using 10% PAGE.

To measure BZU3 enzymatic kinetics, 25 nM BZU3 was incubated with different concentrations of UDP-Glc or UDP-Gal dissolved in 10 mM Tris-HCl (pH 8.0) buffer at 30 °C for 1 min. To measure the ratio between UDP-Glc and UDP-Gal after the reactions reached equilibrium, 2 μM BZU3 was incubated with 2 mM of UDP-Glc or UDP-Gal in 10 mM Tris-HCl (pH8.0) at 30 °C for 30 min. The reactions were stopped at 70 °C for 10 min. After the reaction solutions were filtered with a 0.45 μm filter, the surplus of UDP-Glc and products of UDP-Gal were measured using HPLC. Similar procedures were performed with BZU3$^{S133A}$ and BZU3$^{Y157F}$ to examine the effects of mutation at the active sites. Catalysis between UDP-GlcNAc and UDP-GalNAc was followed using a similar protocol.

## Cellulose staining

To check the cellulose content in the third leaf base of 7-day-old seedlings, the tissues were decolorized in 12.5% acetic acid for 2 h before being dehydrated in 100% and 50% ethanol for 4-5 h. After rinsing in ddH$_2$O for 20 min, the samples were stained with 0.02% (w/v) Direct Red 23 for 5 h and then rinsed with dd H$_2$O. The samples were observed using a Zeiss LSM710 confocal microscope with excitation at 561 nm. The method is based on that of Riglet et al.[31].

## Immunofluorescence analysis

Mixed-linkage glucans were detected by immunofluorescence labeling using methods adapted from Giannoutsou et al.[47]. Briefly, the 2-3 cm leaf base of WT and *bzu3* mutant plants were cut into thin strips, 1-

2 mm wide and 3-5 mm long, and were fixed in 2% (v/v) paraformaldehyde and 0.1% (v/v) glutaraldehyde in PEM buffer (pH 6.8) for 1.5 h at 4 °C. After rinsing in PEM buffer, the fixed samples were dehydrated in gradient ethanol series and post-fixed in 0.25% (w/v) osmium tetroxide. The samples were infiltrated and embedded with LR White acrylic resin (LRW, Sigma-Aldrich). Sections (1 μm thickness) were transferred onto poly-lysine glass slides and were blocked with 5% BSA (v/v) in 1×PBS solution for 5 h. To detect the mixed-linkage glucans, BG-1 primary antibodies (Biosupplies, Australia) were diluted 1:100 in 1×PBS buffer containing 2% BSA, and were incubated with the samples at 4 °C overnight. After rinsing three times with 1×PBS buffer, the samples were incubated with secondary antibodies (anti-mouse IgGs conjugated with FITC) diluted 1:500 in the same buffer at 37 °C for 1 h. After rinsing a further three times, the slides were mounted with anti-fade medium containing p-phenylenediamine, and were observed using a confocal microscope (Zeiss, LSM710) with excitation at 561 nm.

### Stimulated Raman scattering microscopy (SRS)

Label-free imaging of cellulose and hemicellulose was performed with a home-built stimulated Raman scattering microscope[48]. A tunable picosecond two-color laser (picoEmerald-S, APE GmbH, Germany) served as the light source. It generated two synchronized 80 MHz picosecond laser beams. The 1032 nm Stokes beam was modulated at 20 MHz. The pump beam was tuned to 926.5 nm to detect cellulose and hemicellulose. Stokes and pump beams were coupled into a laser-scanning upright microscope (FVMPE, Olympus, Japan) equipped with a 25X water-immersion objective lens (XLPLN25XWMP2, Olympus, Japan). SRS signals were collected by a photodiode and lock-in amplifier module (customized from APE GmbH, Germany). Fresh maize leaf epidermal strips were placed on a glass slide with water and covered with a cover glass. SRS images (512 × 512 pixels) were acquired. The pixel dwell time was 2 to 4 μs. Each image was line-averaged 3 to 5 times to reduce noise. The signal intensity corresponds to the content of cellulose and hemicellulose.

### UPLC-MS analysis of endogenous UDP-Glc and UDP-Gal contents

The third leaf base of 7-day-old seedlings (Elongation stage of GC morphogenesis) was collected and ground to a fine powder in liquid nitrogen. $100 \pm 2$ mg of plant materials was weighed and extracted with 1 mL of pre-chilled methanol/water (3/1, v/v) including 0.1% formic acid. The samples were vortexed for 1–2 min and sonicated for 15 min at room temperature before being centrifuged at 8,500 g for 10 min at 4 °C. The supernatant was filtered using a 0.22 μm membrane and transferred to glass vials for analysis.

UDP-Glc and UDP-Gal were quantified by ultra-high-performance liquid chromatography-tandem mass spectrometry (UPLC-MS/MS). Samples were analyzed using a Xevo TQ-XS system (Waters, USA) with an ESI ion source. The data were collected under the negative ion mode and multiple reaction monitoring mode. Precursor and fragment ions are: UDP-Glc (m/z 564.8–322.9) and UDP-Gal (m/z 564.8–322.9). Chromatographic separation was conducted with a BEH Amide column (2.1 × 100 mm, 1.7 μm) maintained at 30 °C. The mobile phase includes solvent A (50 mM ammonium formate, adjusted to pH 3.6 with formic acid) and solvent B (acetonitrile), with a flow rate of 0.4 ml·min⁻¹. The isometric gradient system was set to 0–35 min, 21% solvent B. The autosampler temperature was set to 4 °C and 5 μL of the sample was injected.

### Subcellular localization of BZU3

To observe the expression pattern of BZU3, *BZU3pro:BZU3-YFP,* and the nuclear marker *35S:H2B-mCherry* were transformed transiently into maize protoplasts. The localization of the proteins was observed by confocal microscopy (Zeiss, LSM710). The methods were based on Zhang et al.[49]. Maize protoplasts were isolated from a 21-day-old etiolated seedling of inbred line B73 as previously described. 10 μg of each plasmid was used and the transformed protoplasts were incubated overnight.

### Bimolecular fluorescence complementation assays

The coding sequences of *BZU3* and *BZU3^bzu3-2^* were amplified from the genome of the wild-type Zheng58 and mutant *bzu3-2* plants. Constructs of *35S:BZU3-cYFP*, *35S:nYFP-BZU3*, *35S:BZU3^bzu3-2^-cYFP*, *35S:nYFP-BZU3^bzu3-2^* were respectively made. The primers were listed in Supplementary Table S7. Corresponding pairs of plasmids were transformed into tobacco cells using *Agrobacterium tumefaciens* GV3101. Two or three days after inoculation, fluorescence patterns were observed using a confocal microscope (Zeiss LSM710).

### Crystallization and structure determination

The purified BZU3 protein (~ 5.0 mg/ml) was crystallized using the sitting-drop vapor diffusion method against a reservoir solution containing 100 mM sodium malonate, pH 5.0 and 12 - 16% (w/v) PEG3350. To prepare the enzyme complex bound with UDP-Glc or UDP-GlcNAc, the long-rod shape BZU3 crystals were soaked by the addition of 5 mM UDP-Glc or UDP-GlcNAc in their mother-liquors overnight before harvest. Cryoprotection was obtained by the sequential addition of mother-liquors supplemented with 25% (w/v) glycerol, followed by flash freezing in liquid N₂. The X-ray diffraction datasets were collected at the beamline BL19U1 of the Shanghai Synchrotron Radiation Facility (SSRF). Data processing and reduction were carried out using the XDS package[50]. The structures were solved by molecular replacement using Phaser-MR using human GALE (PDB 1HZJ, sharing 58% sequence identity) as the search model[51]. The models were completed using the COOT program, and further refinement was done with the Phenix package[52]. Structure validation was performed with Molprobity[53]. Details for crystallization, data collection, structure refinement, and analyses are shown in Supplemental Table S5. The pictures shown in the paper were prepared using PyMOL[54].

### UDP-Glucose treatment assay

The old coleoptile and outer first leaf of *bzu3* mutants (3-day-old seedlings) were carefully removed, leaving the second leaf supported by a toothpick. The leaf base was painted with 450 μM UDP-glucose in 0.1% Triton X-100 solution. The treatment was repeated 8 times/day for 3 days and the treated part was marked. After another 2-3 days when the leaf sheath was formed, the treated leaf bases were collected for observation.

### Glycomics analysis

A total of 1 g samples of maize seedling leaf bases were collected from the wild-type plants and the mutants. The samples were ground in liquid nitrogen, and then dissolved in 2 mL 8 M urea followed by protein alkylation and digestion. The enriched intact N-glycopeptides were then isotopically labeled with diethyl. The samples of each group were mixed in a 1:1 ratio and analyzed online by C18-RPLC-MS/MS[55], as follows: the enriched N-glycome samples were injected into a C18 precolumn (Agilent ZORBAX 300SB, 360 μm o.d. × 200 μm i.d., 7 cm long) for sample loading. Chromatographic separation was performed using a 75 cm-long analytical column (360 μm o.d. × 75 μm i.d.) packed with the same C18 precolumn. The buffers were as follows: Buffer A contains 99.9% H₂O and 0.1% FA, and Buffer B contains of 99.9% ACN and 0.1% FA. The flow rate of the mobile phase was 0.2 mL/min, with a multi-step gradient: 1% B 10 min, 1 - 25% B 20 min, 25 - 60% B 190 min, 60 - 95% B 10 min, 95 - 95% B 10 min, and 95 - 1% B 5 min. MS spectra were detected as follows: m/z range 500 - 2500, mass resolution 35 k, automatic gain control (AGC) target $5 \times 10^5$, max ion injection time 200 ms. MS/MS spectra were obtained using the Top20 data-dependent mode, with the following settings: mass resolution 17.5 k, AGC target $1 \times 10^5$, max ion injection time 200 ms, dynamic exclusion 50 s, HCD normalized collision energy 10%, isolation window 3 Th. The ESI conditions were as follows: spray voltage 2.8 kV, capillary temperature 250 °C, and S-lens RF level 75. Based on the libraries of the Corn protein group and of the plant N-glycosylation modifications in

the Uniprot database, we identified 2194 intact N-glycopeptides in qualitative analysis and quantified 181 differentially expressed intact N-glycopeptides (DEGPs) (two out of three technical replicates, ≥ 1.5-fold change and *p*-value < 0.05). Gene ontology enrichment analysis was conducted using the online tool ShinyGO (http://bioinformatics. sdstate.edu/go/) with a FDR adjusted P-value cutoff of 0.05.

## Verification of N-Glycosylation site
The full-length sequence of *PAN2*, *GT14*, and *FUT* was amplified from maize cDNAs and inserted into the pCAMBIA2300 vector (*35S:PAN2-GFP*, *35S:GT14-GFP*, and *35S:FUT-GFP*). *35S:PAN2^{N554Q}-GFP*, *35S:GT14^{N283Q}-GFP*, and *35S:FUT^{N121Q}-GFP* were constructed using site-directed mutagenesis. The primers were listed in Supplementary Table S7. The plasmids were transformed into *Agrobacterium tumefaciens* GV3101. Subsequently, *35S:PAN2-GFP and 35S:PAN2^{N554Q}-GFP*, *35S:GT14-GFP and 35S:GT14^{N283Q}-GFP*, *35S:FUT-GFP and 35S:FUT^{N121Q}-GFP* were expressed transiently in tabacco leaves. Finally, the total proteins were extracted and analyzed by Western blotting, using anti-GFP antibodies (Transgen, HT801, diluted 1:10000).

## Statistics and reproducibility
The maize plants used in this study were grown in the greenhouse at 16 h/8 h and 32 °C/25 °C (day/night). The samples were obtained from 7-day-old seedlings unless noted otherwise. For analysis, plants of the same genotype were randomly picked for experiments to avoid bias. Sufficient sample sizes were determined by pilot experiments as indicated in the main text, in the figures, and in the figure legends. No data were excluded from the study. All experiments were repeated at least three times, with at least three independent biological replicates.

## Reporting summary
Further information on research design is available in the Nature Portfolio Reporting Summary linked to this article.

## Data availability
The coordinates of the atomic model of UDP-Glc/GlcNAc 4-Epimerase in this paper have been deposited in the Protein Data Bank (PDB). The accession codes for UDP-Glc/GlcNAc 4-Epimerase with NAD, UDP-Glc/GlcNAc 4-Epimerase with NAD/UDP-Glc, and UDP-Glc/GlcNAc 4-Epimerase with NAD/UDP-GlcNAc are 7XPP, 7XPO, and 7XPQ, respectively. The data of glycomics analysis are available via ProteomeXchange with identifier PXD041642. Source data are provided with this paper.

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

## Acknowledgements

We thank Prof. José Ramón Botella, Changqing Zhang, Zongwei Cai, Guiling Sun and Dr. Yue Rui for valuable suggestions. We would like to thank Ms. Lingyu Mi, Qi Zheng and Ruiting Sun at the electron micro-scope platform of State Key Laboratory of Crop Stress Adaptation and Improvement, Henan University. This work was supported by the National Natural Science Foundation of China (U21A20206 and 31970808) and the Program for Innovative Research Team (in Science and Technology) at University of Henan Province (21IRTSTHN019). D.W.G. was supported by grants from the USDA to the College of Agri-culture and Life Sciences at the University of Arizona.

## Author contributions

C.-P.S and S.G. designed and supervised this study. Y.Z., S.G., and C.-P.S. designed the experiment; Y.Z., J.X., X.W. (Xiaohang Wang), X.Q., G.W., W.L., Zu.Li., Y.S., J.W., Zh.Li., Y.N., J.Z., C.L., Z.H., X.Z., S.B., and B.L. performed the experiments and data analysis; Xi.Wa., W.Wu., C.Z., W.Wa., and Y.C. performed crystallization and structure analysis; T.Z., S.G., D.W.G., and C.-P.S. prepared the manuscript with the inputs from other authors. All authors have read, edited, and approved the content of the manuscript.

## Competing interests

The authors declare no competing interests.
