## [Peer Review File · Nature Communications]

REVIEWER COMMENTS

Reviewer #1 (Remarks to the Author):

To study the role of cell wall metabolism for plant growth and development is crucial for a comprehensive understanding of plant life. The formally carefully written and illustrated MS makes novel experimental contributions to this field, however, the experimental design and explanatory model is flawed.

The *bzu3* mutant in maize has a defect in late guard cell development suggesting abnormal cell wall structure. *BZU3* is shown to be a UGE-like gene that encodes for an enzyme that efficiently interconverts UDP-Glc/UDP-Gal as well as the respective UDP-N-acetyl-amino-hexoses. Apart from a thorough enzymatic analysis the MS also presents a crystal structure of *BZU3* and some experiments on its homo-dimeric structure. The mutant phenotype is characterized in a variety of ways including microscopic analysis and proteomics of N-glycosylated proteins. Taken together the MS presents a large amount of experimental analyses. However, the main conclusion on the *in vivo* function of *BZU3* is inconsistent not only with the existing view on carbohydrate metabolism but also with fundamental received wisdom in biochemistry, which I will outline further below.

Previously, four types of plant UDP-sugar 4 epimerases are known UGEs that specifically interconvert UDP-Glc/UDP-Gal, UGEs that moreover interconvert UDP-Xyl/UDP-L-Ara, as well as GAEs and UGEs that interconvert UDP-hexuronic acids and UDP-pentoses, respectively. To my knowledge UDP-GlcNAc/UDP-GalNAc were no substrates for plant UGE-like enzymes (e.g. Kotake et al 2009 Biochem J. 424:169ff). The MS shows that the conversion rate of UDP-GlcNAc/UDP-GalNAc is comparable to the conversion rate of UDP-Glc/UDP-Gal adding a new activity to plant UGEs and also giving a possible biochemical solution to the controversial existence in plants of UDP-GalNAc. This observation is particularly interesting given the presentation of a crystal structure for *BCU3*. Previously, based on human UGE, the *E. coli* UGE was mutated to accommodate the additional volume of UDP-GlcNAc/UDP-GalNAc in substrate binding pocket (Thoden et al 2002 JBC 277: 27528ff) and it would be interesting to see how the present structure fits with the previous work. It also would be interesting if *BCU3* also interconverts UDP-Xyl/UDP-L-Ara like *AtUGE1* and -3 (Kotake Ref above) because both sugars are prominent in cell wall polysaccharides of grasses. On the other hand, the UDP-GlcNAc/UDP-GalNAc activity is enigmatic as to its metabolic role. The biosynthesis of UDP-GlcNAc is thought to proceed via the intermediate N-acetylglucosamine-6-P (see e.g. Bar Peled 2011 Ann Rev Plant Biol 62:127ff) while the biosynthesis of UDP-GalNAc wherever it is known follows the 4-epimerization of UDP-GlcNAc. So the pathway that is described in the manuscript clearly begs the question where the UDP-Gal and the UDP-GalNAc could come from if not from an enzyme with the same activity as *BZU3*.

In my opinion the first lapse of judgement is to conclude from the enzyme-kinetic properties of *BZU3* to the direction of metabolite flux *in vivo*. It is stated in line 257 that "Comparing these K_{cat}/K_m values suggests that *BZU3* prefers UDP-Gal or UDP-GalNAc as substrates during the catalysis of the isomerization." in biochemical terms this is nonsensical because the ratio of the K_{cat}/K_m values for the reactants on both sides of an equation is defined by the equilibrium constant (as stated by Haldane's equation). The equilibrium on the other hand is a thermodynamic constant that is not influenced by enzyme kinetic properties at all. That a catalyzer does not determine the direction of a reaction but only its velocity is high school knowledge, but it is not followed by the authors.

The introduction of mutations in *BZU3* that render the protein enzymatically inactive is unsurprising given the conserved nature of the chosen residues and much less space should be dedicated to this experiment that gives no new insight.

This assumption that *BZU3* mediates flux from UDP-Gal to UDP-Glc *in vivo*, appears to put a bias on subsequent experiments. For instance it is certainly a good idea to try to chemically compensate the mutant defect but why was only UDP-Glc used? The most obvious sugar to use for complementation is galactose as its biosynthesis solely depends on UGE. Galactose feeding has for instance, restored the *uge4* mutant in *Arabidopsis* (Seifert et al 2002 Curr Biol 12:1840ff). While uptake channels and metabolic salvage pathways are well known for D-Glc and D-Gal it is unlikely that the highly charged UDP-Glc would actually enter the cell (this is a well-known problem in the design of nucleotide sugar based inhibitors). The concentration (<0.5mM) of UDP-Glc that was used in this experiment is

probably below the intracellular UDP-Glc concentration so even if there was a channel the treatment would be unlikely to accumulate higher levels.

In this respect it should also be said that the analysis of UDP-Glc/UDP-Gal levels was done in explants that are likely to express additional UGEs and contain many cell types where BZU3 isn't even expressed, which makes it difficult to explain the observed differences but which is not mentioned in the manuscript.

The analysis of the cell wall has only extended to glucans but has not looked at galactans, arabinogalactans or other cell wall matrix polysaccharides.

What is also surprising is that the authors never given an explanation on the source of UDP-Gal that is suggested to be converted to UDP-Glc by BZU3.

In summary while some data are interesting (mutant characterization, BZU3 expression and activity, and BZU3 structure), other experiments are either out of context e.g. (N-glycoproteomics), vesicle dynamics, or flawed (UDP-Glc feeding, nucleotide sugar analysis) and other experiments are necessary (cell wall carbohydrate analysis, rescue experiments with hexoses such as D-Gal L-Ara, D-Glc). For me it is difficult to make specific recommendations on improvements other than to completely re-draft the manuscript in a less biased way.

Reviewer #2 (Remarks to the Author):

The authors collected cellular, biochemical, and structural data to show that BZU3, a maize UDP-glucose 4-epimerase, plays a key role in modulating stomatal morphogenesis. Mutants (plants, genes, and proteins) were used as a tool to study gene function and stomatal development. In vitro enzymatic assays and crystallographic images indicate that BZU3 catalyzes conversion of UDP-Gal/GalNAc to UDP-Glc/GlcNAc. N-glycoproteomics analysis. This study is quite comprehensive. No major flaws were found. The information described is a good contribution to the literature. It is recommended to accept the manuscript after major revisions. However, there are a few comments for the authors to consider.

Major:

The UPLC-MS/MS conditions the analysis of UDP-Glc and UDP-Gal were described (L655-675). However, the method of N-glycopeptide analysis (L723, glycomics analysis) was inadequately described, which needs to be further detailed.

In general, N-glycosylation involves the attachment of a glycan to Asn through the recognition of a sequence motif called a "sequon" i.e. Asn-X-Ser/Thr, where X is any amino acid except for proline. How many N-glycosylation sites are on BZU-3?

It is much easier to use LC/MS analysis to confirm the glycosylation sites than the gel shift assay of proteins generated by site directed mutagenesis.

Minor:

L34: immunofluorescence intensity of ... was reduced ...

L113-116: this part may belong to the introduction and additional information of the maize mutant bizui3 (bzu3) may be added in the introduction.

L268, L271 and other places in the manuscript: was significantly. Please indicate p value

L338: This sentence may need revision. BZU3 catalyzes the conversion of UDP-Glc to UDP-Gal, while a reduced level of UDP-Glc led to the stomatal phenotype in bzu3 (L307), i.e., effects on morphogenesis of maize GCs.

L375-379: To confirm the glycosylation sites of the proteins, LC/MS analysis is much more convincing and direct than the gel shift assay of proteins generated by site directed mutagenesis.

L472: Should it be the glycosylation site rather sites since you are talking about N554?

Reviewer #3 (Remarks to the Author):

Comments

Zhou et al. identified a novel *bziou* (*bzu*) mutant (*bzu3*), which affects the differentiation and morphogenesis of the dumbbell-shaped GCs in maize. The mutation is mapped to a maize UDP-glucose 4-epimerase that regulates UDB-glucose and UDP-NGlcNac levels. Additional alleles and transgenic complementation confirm that this epimerase in fact constitutes BZU3. Careful in-vitro enzymatic assays and structural biology approaches show that BZU3 indeed regulates UDP-Glc levels. The morphogenetic defects were linked to cell wall thickening and cell wall biogenesis defects in the central part of the dumbbell-shaped GCs around the stomatal pore. In addition, N-glycosylation is impaired suggesting additional pathways to be defective in addition to cell wall synthesis/maturation.

This is a well-written manuscript, with well-designed and well-executed experiments and an impressive amount of data (e.g. the many maize transgenics reported). The methodology is sound and the conclusions are well supported by the data presented. In addition, I want to applaud the authors to include segregation analysis in Table S1-S3, which I as a geneticist highly appreciate. I have no major concern, only a few intermediate and minor concerns.

Intermediate concerns:

1. Developmental defects and cell wall maturation

The circumstantial evidence is very strong that cell wall formation and consequently cell wall thickening is affected in *bzu3* mutants. An alternative hypothesis is, however, that a key developmental step prior to extensive cell wall modification, maturation and thickening is affected and that the cell wall defects are simply downstream steps that are not occurring due to the earlier developmental arrest. Therefore, I would like to encourage the authors to provide a more detailed spatiotemporal analysis of GC morphogenesis using 3D stacks rather than just simple single-plane images as provided in Figure 1e. This will help determine for example if the pore is always properly formed, which is not easy to determine from single-plane images. In addition, this possibility should be mentioned and discussed.

2. Defects in vesicular trafficking

I do not understand why this data was included in this paper. First of all, not enough methodological details are provided. Second, if indeed cell wall biogenesis and thickening is the major developmental defect, the vesicle number and size could simply be different due to a lack of cell wall biosynthesis components that can be trafficked and/or a consequence of dying/aborting GCs. Therefore, this is likely only a secondary effect. So either the authors significantly strengthen this aspect and use pharmacological treatments to show whether indeed secretory pathway is affected (i.e. by blocking exocytosis) or whether general recycling is affected (by also blocking endocytosis). Also, much more details must be provided on how the vesicle number and size are counted. Alternatively, this data can be removed w/o affecting the conclusions of this work at all.

3. Cell-type specific complementation

Does the GC-specific complementation (and the endogenous promoter construct, by the way) rescue whole plant growth phenotype? Please provide images of the growth habitus and some simple quantification regarding this. This would show that BZU3 is indeed solely required in the stomatal lineage. In addition, on line 447, the authors say that BZU3 "specifically functions" in stomata. I am not sure if this is true. I would rather say, "is specifically required".

4. PAN2 related findings

Sutimantanapi et al. (2014) found a much stronger defect for a pan2 mutant than what is shown in Fig. S16 for bzu3. While a KO could very well affect this process much stronger than bzu3, where N-glycosylation of PAN2 is impaired, the conclusions of the authors are that the phenotype is similar (line 484ff). Without actually showing this in their growing conditions and with their quantification method, this conclusion is not supported. Please either repeat this analysis together with the pan2 mutant or weaken and discuss this discrepancy.

Minor concerns

5. References

Some key references are missing in the introduction:

Line 64 - statement regarding grass stomatal morphology - eg. Stebbins and Shah 1960 Dev Biol, Rudall et al. 2017 Am J Bot, Nunes et al. 2020, Plant J.

Line 67 - stomatal gas exchange in grasses - please add McAusland et al. 2016 New Phytol

Line 71 & line 431 - GC deformation / cell wall an stomata - please add the modeling papers from Andy Fleming's lab.

Line 75 - dumbbell GC morphogenesis - please add Spiegelhalder & Raissig, 2021, Curr Op Plant Biol

6. Strong statements regarding breeding and plant biotech

While I also think that we should keep the big picture in mind, having such strong statements regarding breeding and biotech is not helpful in a paper that untangles fundamental mechanisms of grass stomatal development. Particularly in the abstract and the intro on line 83ff, these statements should be removed. In addition, line 82ff, states that impeding the breeding of water-use efficient crops is caused because we do not understand GC cell wall biosynthesis. There is a plethora of other aspects impeding these efforts (most aspects of grass stomatal biology, vascular conductance, root biology, cuticular biology, seed to vegetative biomass ratio, etc etc.). I like these aspects being discussed at the end of the discussion (line 496ff) but certainly not in the abstract and introduction.

7. Incomplete glycoprotein Table S7

While I understand that data like the differentially affected glycoproteins are important datasets for the future of the lab, I don't appreciate submitting incomplete Supplementary Tables like Table S7 that only show a fraction of the 181 proteins that are differentially affected in bzu3. I assume that this is an honest mistake, which is why I am ignoring this, but I usually insist on full access to the data before reviewing a paper. Alternatively, I expect at least an explanation and notification that this is an incomplete dataset without trying to sneak this past peer review.

8. Additional minor points

Line 127 - Please refer to Fig. S1C.

Line 211, Fig. 2, Fig. S6 - please use ZmMUTE instead of BZU2. People not familiar with the authors previous work might not know that BZU2 is MUTE. I do like naming new genes without a well-established name (like BZU3), but this is not the case for MUTE.

Fig. S1e - I am not sure what the scatter in the graphs mean. It says 10 individuals but many more scatters are indicated. Please clarify or change.

Fig. S16a-c. Please provide original blots.

Line 64 - SCs are not unique to graminoid GCs (see Rudall et al. 2013)

Line 66 - what do you mean by photosynthetic efficiency ? is this a thing like WUE?

Line 372 - please rephrase "and so on"

Point-to-point response to reviewer's comments

Reviewer #1 (Remarks to the Author):

To study the role of cell wall metabolism for plant growth and development is crucial for a comprehensive understanding of plant life. The formally carefully written and illustrated MS makes novel experimental contributions to this field, however, the experimental design and explanatory model is flawed.

*The *bzu3* mutant in maize has a defect in late guard cell development suggesting abnormal cell wall structure. *BZU3* is shown to be a UGE-like gene that encodes for an enzyme that efficiently interconverts UDP-Glc/UDP-Gal as well as the respective UDP-N-acetyl-aminohexoses. Apart from a thorough enzymatic analysis the MS also presents a crystalstructure of *BZU3* and some experiments on its homo-dimeric structure. The mutant phenotype is characterized in a variety of ways including microscopic analysis and proteomics of N-glycosylated proteins. Taken together the MS presents a large amount of experimental analyses. However, the main conclusion on the *in vivo* function of *BZU3* is inconsistent not only with the existing view on carbohydrate metabolism but also with fundamental received wisdom in biochemistry, which I will outline further below.*

*1. Previously, four types of plant UDP-sugar 4 epimerases are known UGEs that specifically interconvert UDP-Glc/UDP-Gal, UGEs that moreover interconvert UDP-Xyl/UDP-L-Ara, as well as GAEs and UXEs that interconvert UDP-hexuronic acids and UDP-pentoses, respectively. To my knowledge UDP-GlcNAc/UDP-GalNAc were no substrates for plant UGE-like enzymes (e.g. Kotake et al 2009 Biochem J. 424:169ff). The MS shows that the conversion rate of UDP-GlcNAc/UDP-GalNAc is comparable to the conversion rate of UDP-Glc/UDP-Gal adding a new activity to plant UGEs and also giving a possible biochemical solution to the controversial existence in plants of UDP-GalNAc. This observation is particularly interesting given the presentation of a crystal structure for *BCU3*. Previously, based on human UGE, the *E.coli* UGE was mutated to accommodate the additional volume of UDP-GlcNAc/UDP-GalNAc in substrate binding pocket (Thoden et al 2002 JBC 277: 27528ff) and it would be interesting to see how the present structure fits with the previous work. It also would be interesting if *BCU3* also interconverts UDP-Xyl/UDP-L-Ara like *AtUGE1* and -3*

(Kotake Ref above) because both sugars are prominent in cell wall polysaccharides of grasses. On the other hand, the UDP-GlcNAc/UDP-GalNAc activity is enigmatic as to its metabolic role. The biosynthesis of UDP-GlcNAc is thought to be proceed via the intermediate *N*-acetylglucosamine-6-*P* (see e.g. Bar Peled 2011 *Ann Rev Plant Biol* 62:127ff) while the biosynthesis of UDP-GalNAc wherever it is known follows the 4-epimerization of UDP-GlcNAc. So the pathway that is described in the manuscript clearly begs the question where the UDP-Gal and the UDP-GalNAc could come from if not from an enzyme with the same activity as BZU3.

Answer: Thanks for your summary on the basis for the metabolism of nucleotide sugars in plants and comments on our manuscript. Firstly, we always pay attention to the debate if UDP-GlcNAc/UDP-GalNAc were substrates for plant UGE-like enzymes. Our biochemical and structural results indicated that BZU3 can catalyze the conversion rate of UDP-GlcNAc/UDP-GalNAc. These data update the previous research and affirm more functions of UGEs, and this is one of the novel points of our research. Following the comments, we compared the structure difference between BZU3 and human UGE in the previous work in the text. In Thoden's work, the mutation of Y299C in *E.coli* UGE resulted in a gaining of UDP-GalNAc activity by more than 230-fold. Similar to the Y299C mutation in *E. coli* UGE, the corresponding position in BZU3 is valine, both Y to C or V at this position provide the additional volume of UDP-GlcNAc in substrate binding pocket (Figure 1). This structure similarity may confer BZU3 the ability to catalyze the UDP-GlcNAc/UDP-GalNAc conversion.

Figure 1. Superposition of the UDP-GlcNAc binding pocket for the BZU3 and *E. coli* UGE.

Secondly, we thank the reviewer's suggestion to analyze if BZU3 has the enzyme activities for the interconversion between UDP-Xyl and UDP-L-Ara. As shown in the supplementary Fig. S14. BZU3 can indeed convert about 45% of UDP-Xyl to UDP-L-Ara, but almost cannot convert UDP-L-Ara to UDP-Xyl under our experimental conditions. This function of BZU3 is consistent with the more xylan in grass guard cell walls compared with dicot.

Thirdly, it is a good question where are the sources of UDP-Gal and the UDP-GalNAc in the stomatal model system. As we know, plants utilize the various nucleotide sugars in many ways to build glycans, including the carbohydrate derived from photosynthesis, the sugar generated by hydrolyzing translocated sucrose, the sugars released from storage carbohydrates, the salvage of sugars from glycoproteins and glycolipids, the recycling of sugars released during primary and secondary cell wall restructuring, and the sugar generated during plant-microbe interactions (Bar-Peled and O'Neill, 2011 Ann Rev Plant Biol 62:127). One possible source of UDP-Gal and the UDP-GalNAc in stomata is to recycle the free sugars from the turnover of glycans during development via the salvage pathways. For example, in the presence of SLOPPY and UTP (Yang et al., 2015, J Biol Chem 284: 21526), α -D-Gal-1-P and 2-deoxy- α -D-Gal-1-P can be converted to UDP- α -D-Gal and UDP-2-deoxy- α -D-Gal through the Arabidopsis galactokinase, respectively. Although there is no biochemical evidence that GalK (GALACTOSE KINASE 1) exists in maize, our genomic survey indicated that there are an ortholog gene (Zm00001d034733) located in the genomics of maize. In addition, we first reported that BZU3 as an isomer specifically functions in guard cells in a single-cell system. We suspect that a large amount of UDP-Glc is needed at a specific time during the cell wall synthesis, and thus BZU3 isomerase is needed to provide sufficient UDP-Glc.

2. In my opinion the first lapse of judgement is to conclude from the enzyme-kinetic properties of BZU3 to the direction of metabolite flux in vivo. It is stated in line 257 that "Comparing these Kcat/Km values suggests that BZU3 prefers UDP-Gal or UDP-GalNAc as substrates during the catalysis of the isomerization." in biochemical terms this is nonsensical because the ratio of the Kcat/Km values for the reactants on both sides of an equation is defined by the equilibrium constant (as stated by Haldane's equation). The equilibrium on the other hand is a thermodynamic constant that is not influenced by enzyme kinetic properties at all. That a catalyzer does not determine the

direction of a reaction but only its velocity is highschool knowledge, but it is not followed by the authors.

Answer: We agree that a catalyzer does not determine the direction of a reaction but only its velocity. We have corrected the inaccurate description. However, the main conclusion on BZU3 function in the formation of the dumbbell-shaped guard cell is valid, because this is mainly based on the fact that cellulose and MLG synthesis were severely impaired, in which glucose is a quantitatively major component of cellulose and MLG. Moreover, we know that one enzyme has more than one potential substrate. The K_{cat}/K_m values can be used to determine the specificity of the enzyme for different substrates. The higher the value, the more specific the enzyme is for the substrate (Lorsch, J. R. 2014, Methods Enzymol., Academic Press. pp 3-15). Because the most suitable substrate will show a high value of K_{cat} and a low value of K_m . This principle is also suitable for UDP-glucose 4-epimerase (Shin, S.-M et al., 2015, Arch. Biochem. Biophys. 585: 39-51). In the new version, we have changed this sentence to ‘BZU3 converted UDP-Gal/UDP-GalNAc to UDP-Glc/UDP-GlcNAc faster in the reversible isomerization reaction’. Please see Lines 256-257.

The introduction of mutations in BZU3 that render the protein enzymatically inactive is unsurprising given the conserved nature of the chosen residues and much less space should be dedicated to this experiment that gives no new insight.

Answer: We agree with the reviewer that experiments of mutations in BZU3 give no new insight. Our purpose of the enzyme mutation introduction and activity assay of S133A and Y157F here is to obtain the mutated BZU3 without catalytic activity but maintaining normal protein structures. This mutation was used for further transforming into *bzu3* mutants. The transgenic plants could not recover the *bzu3* phenotypes, which indicated that the phenotypes of *bzu3* mutants were caused exactly by the loss of UGE activity.

*This assumption that BZU3 mediates flux from UDP-Gal to UDP-Glc in vivo, appears to put a bias on subsequent experiments. For instance it is certainly a good idea to try to chemically compensate the mutant defect but why was only UDP-Glc used? The most obvious sugar to use for complementation is galactose as its biosynthesis solely depends on UGE. Galactose feeding has for instance, restored the *uge4* mutant in Arabidopsis (Seifert et al 2002 Curr Biol 12:1840ff). While uptake channels and*

metabolic salvage pathways are well known for D-Glc and D-Gal it is unlikely that the highly charged UDP-Glc would actually enter the cell (this is a well-known problem in the design of nucleotide sugar based inhibitors). The concentration (<0.5mM) of UDP-Glc that was used in this experiment is probably below the intracellular UDP-Glc concentration so even if there was a channel the treatment would be unlikely to accumulate higher levels.

Answer: Regarding the subsequent experiments to chemically compensate for the mutant defect, the seedlings of *bzu3* mutants were fed with UDP-Glc, UDP-Gal, Glc, Gal, Xyl, and Ara. We only observe a trend towards complementation for the dumbbell-shaped stomata in the *bzu3* mutant by higher concentrations of UDP-Glc feeding. The results of UDP-Glc complementation are consistent with the recent report (Xia et al., 2021, New Phytol.), in which treating leaves with 150 μ M of UDP-Glc will result in corresponding phenotypes. Although there is currently no report on UDP glucose transport proteins or channels, it is likely that UDP-Glc externally may cause a small amount of infiltration into cells.

On the other hand, it is unclear if free sugars in the apoplast could be transported across the plasma membrane. The functional characterization of an Arabidopsis plasma membrane-localized sugar transporter AtPLT5 suggests that plants do have the ability to transport glycoses from the apoplast to the cytosol (Klepek YS et al., 2005, Plant Cell 17:204-218). Therefore, Gal application can recover the phenotype of Arabidopsis mutant *uge4* (Rosti et al., 2007, Plant Cell 19:1565-1579). By contrast, mutant *bzu3* was not recovered under the treatment of Glc, Gal, Xyl, and Ara. The possible reason could be that the mechanisms that regulate UDP-Glc/Gal synthesis differ between cells, tissues, and plant species. The maize stomatal BZU3 can interconvert UDP-Glc/Gal as a specific requirement of monosaccharide during polysaccharide synthesis.

In this respect it should also be said that the analysis of UDP-Glc/UDP-Gal levels was done in explants that are likely to express additional UGEs and contain many cell types where BZU3 isn't even expressed, which makes it difficult to explain the observed differences but which is not mentioned in the manuscript. The analysis of the cell wall has only extended to glucans but has not looked at galactans, arabinogalactans or other cell wall matrix polysaccharides. What is also surprising is that the authors never given an explanation on the source of UDP-Gal that is suggested to be converted to UDP-Glc by BZU3.

Answer: Thank you. We apologize for the confusion. With the current technology, we cannot directly measure the content of UDP-Glc/UDP-Gal in guard cells. Considering the difference in BZU3 activity between wild-type and mutant, we selected the maize leaf base with the highest BZU3 expression to avoid interference caused by the background as much as possible. We compared the UDP-Glc/UDP-Gal content in this section between the wild-type and mutant to reflect the function of BZU3. Our results show that there is indeed a change in UDP-Glc/UDP-Gal content in this material, which is consistent with our speculation.

Except for glucan (cellulose and MLG), we also studied the pectin component of the cell wall at the cellular level through immunofluorescence. The results showed that the content of pectin was low in maize leaves, especially in GC where the content was extremely low, and it was difficult to determine whether there were any differences. Please see supplementary Fig. S15.

As described in the response to the first question, we discuss the sources of UDP-Gal and the UDP-GalNAc in the stomatal model system (please see our response to the first comment). The possible source of UDP-Gal in stomata is to recycle the free sugars from the turnover of glycans during development via the salvage pathways. For example, in the presence of SLOPPY and UTP (Yang et al., J Biol Chem 2015, 284: 21526), α -D-Gal-1-P and 2-deoxy- α -D-Gal-1-P can be converted to UDP- α -D-Gal and UDP-2-deoxy- α -D-Gal through the Arabidopsis galactokinase, respectively. Although there is no biochemical evidence that GalK exists in maize, our genomic survey indicated that there are homolog genes located in the genomics of maize. In addition, we first reported that BZU3 as an isomer specifically functions in guard cells in a single-cell system.

In summary while some data are interesting (mutant characterization, BZU3 expression and activity, and BZU3 structure), other experiments are either out of context e.g. (N-glycoproteomics), vesicle dynamics, or flawed (UDP-Glc feeding, nucleotide sugar analysis) and other experiments are necessary (cell wall carbohydrate analysis, rescue experiments with hexoses such as D-Gal L-Ara, D-Glc). For me it is difficult to make specific recommendations on improvements other than to completely re-draft the manuscript in a less biased way.

Answer: Thanks for your constructive suggestions. We removed results of vesicle dynamics and added new experimental data including carbohydrate analysis and rescue

experiments. Based on all the comments above, we rewrote the corresponding parts of the manuscript.

Reviewer #2 (Remarks to the Author):

The authors collected cellular, biochemical, and structural data to show that BZU3, a maize UDP-glucose 4-epimerase, plays a key role in modulating stomatal morphogenesis. Mutants (plants, genes, and proteins) were used as a tool to study gene function and stomatal development. In vitro enzymatic assays and crystallographic images indicate that BZU3 catalyzes conversion of UDP-Gal/GalNAc to UDP-Glc/GlcNAc. N-glycoproteomics analysis. This study is quite comprehensive. No major flaws were found. The information described is a good contribution to the literature. It is recommended to accept the manuscript after major revisions. However, there are a few comments for the authors to consider.

Major:

The UPLC-MS/MS conditions the analysis of UDP-Glc and UDP-Gal were described (L655-675). However, the method of N-glycopeptide analysis (L723, glycomics analysis) was inadequately described, which needs to be further detailed.

Answer: Thank you. We have provided a detailed description of the methods used for N-glycomics. Please see Lines 725-739.

In general, N-glycosylation involves the attachment of a glycan to Asn through the recognition of a sequence motif called a “sequon” i.e. Asn-X-Ser/Thr, where X is any amino acid except for proline. How many N-glycosylation sites are on BZU-3?

Answer: Thanks for your question, but we are analyzing glycosylation patterns affected by BZU3, not glycosylation of BZU3 itself.

It is much easier to use LC/MS analysis to confirm the glycosylation sites than the gel shift assay of proteins generated by site directed mutagenesis.

Answer: Thank you. We have provided the results of identifying several protein glycosylation sites through RPLC-MS/MS, as shown in Fig. 5d-f. Additionally, we have rephrased the corresponding parts of the manuscript. Please see Lines 381-384.

Minor:

L34: immunofluorescence intensity of ... was reduced ...

Answer: Thank you. We have revised this sentence. Please see Line 33.

L113-116: this part may belong to the introduction and additional information of the maize mutant bizui3 (bzu3) may be added in the introduction.

Answer: Thank you. We have revised this sentence. Please see Lines 98-100, 112.

L268, L271 and other places in the manuscript: was significantly. Please indicate p value

Answer: Thank you. We have revised these sentences. Please see Lines 270, 273, 276, and 312.

L338: This sentence may need revision. BZU3 catalyzes the conversion of UDP-Glc to UDP-Gal, while a reduced level of UDP-Glc led to the stomatal phenotype in bzu3 (L307), i.e., effects on morphogenesis of maize GCs.

Answer: We thank you for the suggestion. However, the mutation Y157F affected not only the conversion between UDP-Glc/UDP-Gal, but also UDP-GlcNAc/UDP-GalNAc. We can only draw the conclusion that the catalytic activity of BZU3 is required for the morphogenesis of maize GCs. We hope the rewording is satisfactory. Please see Line 345.

L375-379: To confirm the glycosylation sites of the proteins, LC/MS analysis is much more convincing and direct than the gel shift assay of proteins generated by site directed mutagenesis.

Answer: We thank the reviewer's suggestion. We have provided the results of identifying several protein glycosylation sites through RPLC-MS/MS, as shown in Fig. 5d-f. Additionally, we have rephrased the corresponding parts of the manuscript. Please see Lines 381-384.

L472: Should it be the glycosylation site rather sites since you are talking about N554?

Answer: Thank you. We have made revision based on your suggestion. Please see Line 475.

Reviewer #3 (Remarks to the Author):

Comments

Zhou et al. identified a novel biziou (bzu) mutant (bzu3), which affects the differentiation and morphogenesis of the dumbbell-shaped GCs in maize. The mutation is mapped to a maize UDP-glucose 4-epimerase that regulates UDP-glucose and UDP-NGlcNac levels. Additional alleles and transgenic complementation confirm that this epimerase in fact constitutes BZU3. Careful in-vitro enzymatic assays and structural biology approaches show that BZU3 indeed regulates UDP-Glc levels. The morphogenetic defects were linked to cell wall thickening and cell wall biogenesis defects in the central part of the dumbbell-shaped GCs around the stomatal pore. In addition, N-glycosylation is impaired suggesting additional pathways to be defective in addition to cell wall synthesis/maturation.

This is a well-written manuscript, with well-designed and well-executed experiments and an impressive amount of data (e.g. the many maize transgenics reported). The methodology is sound and the conclusions are well supported by the data presented. In addition, I want to applaud the authors to include segregation analysis in Table S1-S3, which I as a geneticist highly appreciate. I have no major concern, only a few intermediate and minor concerns.

Intermediate concerns:

1. Developmental defects and cell wall maturation

The circumstantial evidence is very strong that cell wall formation and consequently cell wall thickening is affected in bzu3 mutants. An alternative hypothesis is, however, that a key developmental step prior to extensive cell wall modification, maturation and thickening is affected and that the cell wall defects are simply downstream steps that are not occurring due to the earlier developmental arrest. Therefore, I would like to encourage the authors to provide a more detailed spatiotemporal analysis of GC morphogenesis using 3D stacks rather than just simple single-plane images as provided in Figure 1e. This will help determine for example if the pore is always properly formed,

which is not easy to determine from single-plane images. In addition, this possibility should be mentioned and discussed.

Answer: Thank you. We performed Z-stack imaging and overlapping on the stomatal development process. In the Kidney like stage and the early stage of elongation, the pore formation was normal in the mutant. Please see Figure 2. In addition, we have discussed this possibility in the resubmission version. Please see Lines 472-474.

Figure 2 The 3D stacks images of stomatal staining at different developmental stages in WT and *bzu3-1*. The cell wall was stained with propidium iodide (PI). Scale bar, 10 μ m. The images were obtained from approximately 2-3 cm leaf base of the third leaf of 7-day-old seedlings.

2. Defects in vesicular trafficking

I do not understand why this data was included in this paper. First of all, not enough methodological details are provided. Second, if indeed cell wall biogenesis and thickening is the major developmental defect, the vesicle number and size could simply be different due to a lack of cell wall biosynthesis components that can be trafficked and/or a consequence of dying/aborting GCs. Therefore, this is likely only a secondary effect. So either the authors significantly strengthen this aspect and use pharmacological treatments to show whether indeed secretory pathway is affected (i.e. by blocking exocytosis) or whether general recycling is affected (by also blocking endocytosis). Also, much more details must be provided on how the vesicle number and size are counted. Alternatively, this data can be removed w/o affecting the conclusions

of this work at all.

Answer: Thank you. We agree with your point of view and have deleted these data.

3. Cell-type specific complementation

Does the GC-specific complementation (and the endogenous promoter construct, by the way) rescue whole plant growth phenotype? Please provide images of the growth habitus and some simple quantification regarding this. This would show that BZU3 is indeed solely required in the stomatal lineage. In addition, on line 447, the authors say that BZU3 "specifically functions" in stomata. I am not sure if this is true. I would rather say, "is specifically required".

Answer: Thank you. *BZU3* native and GC specific promoter (*ZmMUTE* and *ZmFAMA*) driving *BZU3* transgenic plants (*BZU3**pro*:*BZU3*-YFP; *bzu3-2*, *ZmMUTE**pro*:*BZU3*-YFP; *bzu3-2* and *ZmFAMA**pro*:YFP-*BZU3*; *bzu3-2*) can restore the entire plant growth phenotype (as shown in Supplementary Fig. S7), when the plants grow 8 and 18-day-old. In addition, we have revised the related content in the resubmission version. Please see Lines 189-191 and 215-217. Meanwhile, we have revise "specifically functions" to "is specifically required" in the reversion. Please see Line 450.

4. PAN2 related findings

Sutimantanapi et al. (2014) found a much stronger defect for a pan2 mutant than what is shown in Fig. S16 for bzu3. While a KO could very well affect this process much stronger than bzu3, where N-glycosylation of PAN2 is impaired, the conclusions of the authors are that the phenotype is similar (line 484ff). Without actually showing this in their growing conditions and with their quantification method, this conclusion is not supported. Please either repeat this analysis together with the pan2 mutant or weaken and discuss this discrepancy.

Answer: Thank you. We have revised our manuscript. Please see Lines 488-491.

Minor concerns

5. References

Some key references are missing in the introduction:

Line 64 - statement regarding grass stomatal morphology - eg. Stebbins and Shah 1960 Dev Biol, Rudall et al. 2017 Am J Bot, Nunes et al. 2020, Plant J.

Line 67 - stomatal gas exchange in grasses - please add McAusland et al. 2016 New

Phytol

Line 71 & line 431 - GC deformation / cell wall an stomata - please add the modeling papers from Andy Fleming's lab.

Line 75 - dumbbell GC morphogenesis - please add Spiegelhalder & Raissig, 2021, Curr Op Plant Biol

Answer: Thank you. We have added related references in the revision. Please see Lines 63, 813-819; 66, 824-825; 70, 431, 826-827; 74, 835-836.

6. Strong statements regarding breeding and plant biotech

While I also think that we should keep the big picture in mind, having such strong statements regarding breeding and biotech is not helpful in a paper that untangles fundamental mechanisms of grass stomatal development. Particularly in the abstract and the intro on line 83ff, these statements should be removed. In addition, line 82ff, states that impeding the breeding of water-use efficient crops is caused because we do not understand GC cell wall biosynthesis. There is a plethora of other aspects impeding these efforts (most aspects of grass stomatal biology, vascular conductance, root biology, cuticular biology, seed to vegetative biomass ratio, etc etc.). I like these aspects being discussed at the end of the discussion (line 496ff) but certainly not in the abstract and introduction.

Answer: Thank you. According to your advice, we have revised the abstract and introduction in the revision. Please see Lines 40-41, 81.

7. Incomplete glycoprotein Table S7

*While I understand that data like the differentially affected glycoproteins are important datasets for the future of the lab, I don't appreciate submitting incomplete Supplementary Tables like Table S7 that only show a fraction of the 181 proteins that are differentially affected in *bzu3*. I assume that this is an honest mistake, which is why I am ignoring this, but I usually insist on full access to the data before reviewing a paper. Alternatively, I expect at least an explanation and notification that this is an incomplete dataset without trying to sneak this past peer review.*

Answer: Thank you for pointing out this mistake. We have updated the data in the revision. Please see Supplementary Table S7. Raw data are available via ProteomeXchange with identifier PXD041642. (Reviewer account details: Username: reviewer_pxd041642@ebi.ac.uk. Password: w6ICkJxV).

8. Additional minor points

Line 127 - Please refer to Fig. S1C.

Line 211, Fig. 2, Fig. S6 - please use ZmMUTE instead of BZU2. People not familiar with the authors previous work might not know that BZU2 is MUTE. I do like naming new genes without a well-established name (like BZU3), but this is not the case for MUTE.

Fig. S1e - I am not sure what the scatter in the graphs mean. It says 10 individuals but many more scatters are indicated. Please clarify or change.

Fig. S16a-c. Please provide original blots.

Line 64 - SCs are not unique to graminoid GCs (see Rudall et al. 2013)

Line 66 - what do you mean by photosynthetic efficiency ? is this a thing like WUE?

Line 372 - please rephrase "and so on"

Answer: Thank you. According to your advice, we have revised all the contents in the revision.

Please see Line 124;

Please see Lines 208-209, Fig. 2, Fig. S6;

Please see annotation of Fig. S1e;

Please see source data;

Please see Line 63;

Please see Line 65;

Please see Line 379.

REVIEWERS' COMMENTS

Reviewer #1 (Remarks to the Author):

my comments were sufficiently addressed

Reviewer #2 (Remarks to the Author):

The authors satisfactorily responded to my comments.

Reviewer #3 (Remarks to the Author):

Thank you for carefully addressing the points criticized. All my concerns were addressed.

Point-to-point response to reviewer's comments

Reviewer #1 (Remarks to the Author):

my comments were sufficiently addressed

Reviewer #2 (Remarks to the Author):

The authors satisfactorily responded to my comments.

Reviewer #3 (Remarks to the Author):

Thank you for carefully addressing the points criticized. All my concerns were addressed.

Answer: We thank all reviewers' comments and suggestions.